# Ehbp1 orchestrates orderly sorting of Wnt/Wingless to the basolateral and apical cell membranes

Yuan Gao [1,2,5], Jing Feng [1,2,5], Yansong Zhang[1,2,3], Mengyuan Yi[1], Lebing Zhang[1,2], Yan Yan[4], Alan Jian Zhu [1,2,3] & Min Liu [1,2]

## Abstract

**Wingless (Wg)/Wnt signaling plays a critical role in both development and adult tissue homeostasis. In the *Drosophila* larval wing disc epithelium, the orderly delivery of Wg/Wnt to the apical and basal cell surfaces is essential for wing development. Here, we identified Ehbp1 as the switch that dictates the direction of Wg/Wnt polarized intracellular transport: the Adaptor Protein complex 1 (AP-1) delivers Wg/Wnt to the basolateral cell surface, and its sequestration by Ehbp1 redirects Wg/Wnt for apical delivery. Genetic analyses showed that Ehbp1 specifically regulates the polarized distribution of Wg/Wnt, a process that depends on the dedicated Wg/Wnt cargo receptor Wntless. Mechanistically, Ehbp1 competes with Wntless for AP-1 binding, thereby preventing the unregulated basolateral Wg/Wnt transport. Reducing *Ehbp1* expression, or removing the coiled-coil motifs within its bMERB domain, leads to basolateral Wg/Wnt accumulation. Importantly, the regulation of polarized Wnt delivery by EHBP1 is conserved in vertebrates. The generality of this switch mechanism for regulating intracellular transport remains to be determined in future studies.**

**Keywords** AP-1; Ehbp1; Wg/Wnt; Wls
**Subject Categories** Cell Adhesion, Polarity & Cytoskeleton; Development; Membranes & Trafficking

## Introduction

The canonical Wingless (Wg)/Wnt signaling, well-known for its evolutionary conservation, is crucial for a variety of biological processes, including cell proliferation, pattern formation, stem cell maintenance, and adult tissue homeostasis (Rim et al, 2022). The proper function of this pathway depends on the accurate intracellular and intercellular transport of Wg/Wnt proteins, along with their precise subcellular localization. While extracellular Wg/Wnt transport has been extensively studied, the mechanisms controlling its intracellular transport in Wg/Wnt-producing cells remain unclear.

The *Drosophila* wing disc, with its well-defined apical-basolateral cellular polarity, serves as an excellent model system for studying polarized morphogen transport, particularly in the context of Wg signaling (Fig. 1A). In addition, the spatial separation of Wg-producing and Wg-receiving cells, which can be differentiated and manipulated using distinct genetic markers, allows for direct observation and analysis of intracellular trafficking processes. This is in contrast to most vertebrate systems, where Wnt-producing and Wnt-receiving cells are intermingled. Thus, the unique characteristics of the *Drosophila* wing disc make it an ideal model for studying Wg intracellular trafficking.

In the epithelium of the *Drosophila* wing imaginal disc, Wg protein synthesis occurs along the dorsal-ventral boundary, with *wg* transcripts produced apically by these cells (Simmonds et al, 2001). Subsequent Wg protein secretion can occur either apically (Chaudhary and Boutros, 2019; Marois et al, 2006) or basolaterally (Marois et al, 2006; Strigini and Cohen, 2000; Wu et al, 2004). Notably, there is a substantial accumulation of extracellular Wg on the basolateral side (Fig. EV1A) (Strigini and Cohen, 2000), suggesting that the primary cellular trafficking destination of Wg protein is the basolateral cell surface. This proposition is supported by basolateral localization of key components in the Wg signaling cascade (Chaudhary et al, 2019; Gallet et al, 2008; McGough et al, 2020; Yamazaki et al, 2016), including the Wg co-receptor Frizzled2, as well as heparan sulfate proteoglycans (HSPGs) Dally and Dally-like (Baeg et al, 2001; Han et al, 2005; Kreuger et al, 2004; Yan et al, 2009), which orchestrate the long-range gradient-dependent signaling of hydrophobic Wg proteins (Herr and Basler, 2012; Liu et al, 2022; Wolf and Boutros, 2023). However, considerable evidence also shows that Wg proteins secreted apically can activate downstream signaling either at apical cell surfaces or in endosomal compartments of signaling receiving cells through apical endocytosis (Chaudhary and Boutros, 2019; Hemalatha et al, 2016; Linnemannstons et al, 2020; Marois et al, 2006). These observations highlight the importance of the apical surface as a pivotal target of intracellular trafficking in Wg-producing cells, strongly suggesting the need to ensure the efficient transport of Wg to both the basolateral and apical cell surfaces for its extracellular distribution and downstream signaling.

[1]Ministry of Education Key Laboratory of Cell Proliferation and Differentiation, School of Life Sciences, Peking University, Beijing 100871, China. [2]Peking-Tsinghua Center for Life Sciences, Academy for Advanced Interdisciplinary Studies, Peking University, Beijing 100871, China. [3]Peking University Chengdu Academy for Advanced Interdisciplinary Biotechnologies, Chengdu, Sichuan 610213, China. [4]Division of Life Science, Hong Kong University of Science and Technology, Clear Water Bay, Kowloon, Hong Kong, China. [5]These authors contributed equally: Yuan Gao, Jing Feng. ✉E-mail: zhua@pku.edu.cn; liumin02@pku.edu.cn

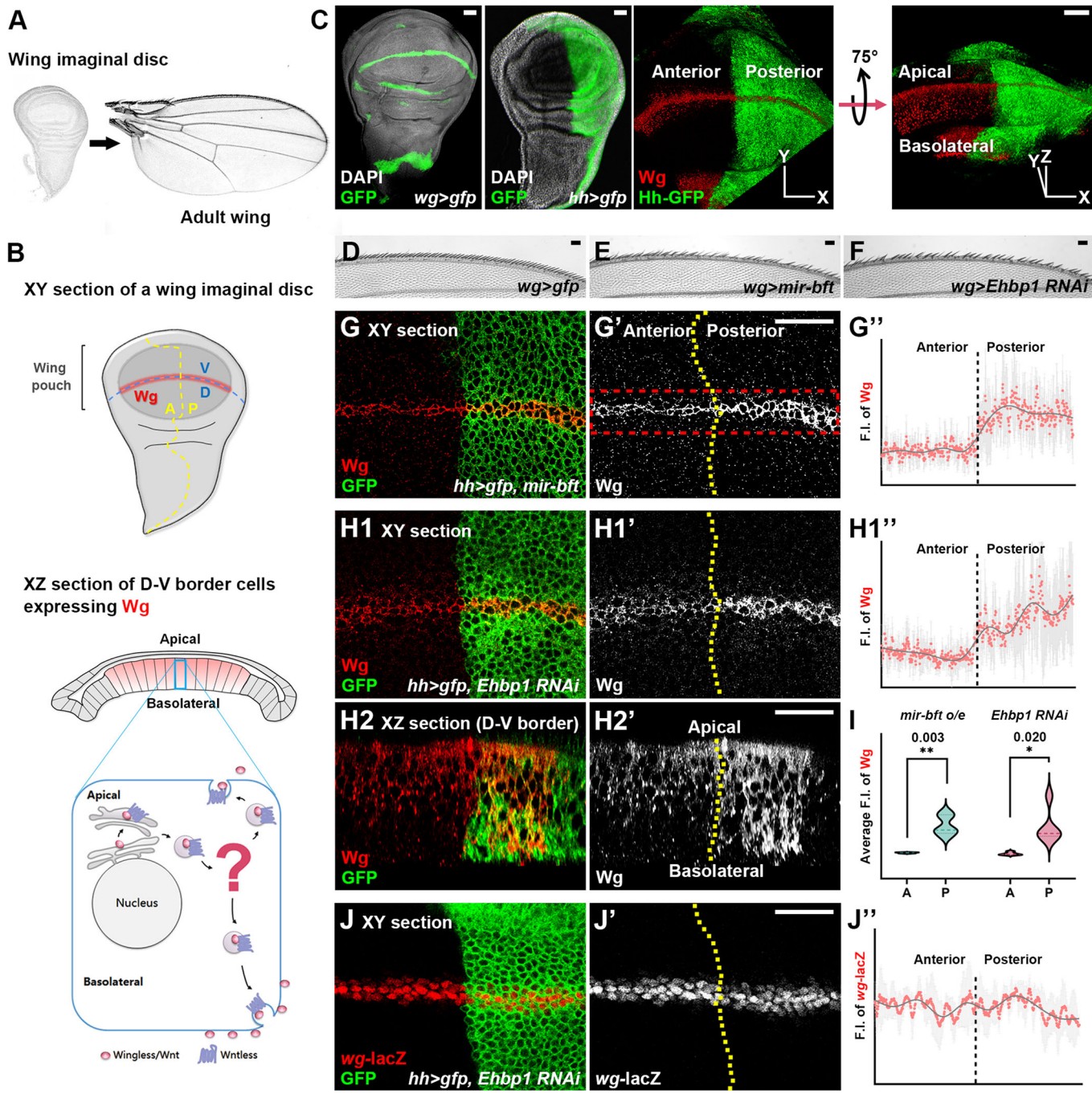

Previous research into the production and trafficking of Wg within distinct membrane compartments has unveiled a transport system that involves apical secretion, followed by transcytosis from the apical to the basolateral extracellular environment (Gallet et al, 2008). Key regulators of this process include components of the exocyst complex, which are known for their essential roles in the apical secretion of Wg (Chaudhary and Boutros, 2019), as well as the E3 ubiquitin ligase Godzilla (Gzl/RNF167), Gzl-targeted Synaptobrevin/VAMP3a, and the kinesin motor Klp98A, all of which are implicated in facilitating Wg transcytosis (Witte et al, 2020; Yamazaki et al, 2016). Disruption of the function of the exocyst complex impedes Wg signaling transduction and wing development. However, attempts to inhibit transcytosis through RNA interference (RNAi) targeting *Klp98A* or *gzl*, or via the introduction of a dominant-negative version of *gzl* (*gzl.LD*), did not result in noticeable anomalies in adult wing margin development (Witte et al, 2020 and Fig. EV1B–I). Furthermore, these interventions exhibited negligible impact on extracellular Wg levels on both apical and basolateral surfaces and did not significantly affect Wg signaling (Witte et al, 2020). These observations are consistent with previous studies indicating that inhibition of *RNF167*, *VAMP3*, or both in polarized vertebrate Madin-Darby canine kidney (MDCK)

**Figure 1.** *Ehbp1* plays a critical role in regulating Wg secretion.

(A, B) The development of the adult wing from the wing imaginal disc in *Drosophila* is a complex and highly orchestrated process. Throughout this progression, cells along the dorsoventral (D-V) boundary respond to high-threshold Wg signaling, which gives rise to the stereotypical morphology of the wing margin. In the *Drosophila* wing disc, cell polarity is distinguished by an apical-basolateral (A-P) organization, creating distinct cellular domains. The apical surface of the cell faces the lumen or adjacent cells in contact, whereas the basolateral surface is located on the cell's sides and base. This polarity is essential for various cellular processes, including morphogenesis and the directional transport of signaling molecules such as Wg, which is typically transported from basolateral to apical or vice versa. The defined apical-basolateral polarity in *Drosophila* wing discs is crucial for tissue organization and the establishment of signaling gradients, supporting proper organ development and patterning. The intracellular trafficking of Wg in the Wg-producing cells of the wing imaginal disc epithelium is schematically summarized. Importantly, a regulatory framework that coordinates the transport of Wg to both basolateral and apical cell domains, maintaining a delicate balance between their respective extracellular Wg gradients, remains to be elucidated. (C) The expression patterns of *wg-Gal4* and *hh-Gal4* in *Drosophila* wing discs are shown. *wg-Gal4* drives the expression of *UAS-gfp* in Wg-producing cells along the D-V boundary, whereas *hh-Gal4* specifically induces *UAS-gfp* expression in the posterior compartment of the wing disc. Images of an *hh >gfp* wing disc are presented, including both an apical view (an XY section) and a dorsolateral view, which is obtained by rotating the XY plane outward for 75 degrees along the horizontal X-axis. (D–F) Overexpression of *mir-bft* (E) or RNAi against *Ehbp1* (F) using the *wg-Gal4* driver led to the loss of sensory bristles on the wing margin. (G–H2') Wg accumulation was observed upon overexpression of *mir-bft* (G–G") or RNAi against *Ehbp1* (H1–H2") using the *hh-Gal4* driver. A 3D reconstruction of the D-V border cells (as viewed in an XZ section) of a wing disc expressing RNAi against *Ehbp1* showed Wg accumulation in both the basolateral and apical domains of the posterior compartments of the wing discs (H2 and H2'). In these and all subsequent figures, dotted yellow lines indicate the A-P boundaries. The plot profiles of immunofluorescence staining were generated within the rectangular areas demarcated by red dashed lines (for each genotype, $n \geq 3$ biological replicates). Data are shown as mean ± SD. An example of this analysis can be seen in (G–G"). (I) The statistical analysis of the Wg immunofluorescence intensity in (G") and (H1") was performed (for each genotype, $n \geq 3$ biological replicates). Data are presented as violin plots. Two-tailed Student's t-tests were employed to analyze the differences between anterior and posterior F.I. *, indicates a significant difference with a *P* value < 0.05; **$P < 0.01$. (J–J") Expression of RNAi against *Ehbp1* by *hh-Gal4* did not alter the expression of *wg-LacZ*. Scale bars, 25 μm. Source data are available online for this figure.

epithelial cells had minimal effects on basolateral WNT1 secretion (Yamamoto et al, 2017). Taken together, these observations suggest that transcytosis in basolateral Wg transport may be dispensable for activating Wg signaling. Is there then an alternative pathway that transports Wnt directly from the trans-Golgi network to the basolateral cell surface?

Here, we reveal a novel regulatory role for *Eps15 homology domain containing protein-binding protein 1* (*Ehbp1*) in the precise sorting of Wg/Wnt proteins to both basolateral and apical cell membranes. Initially identified as interaction partners for Eps15-homology domain-containing proteins EHD1/2, Ehbp1 homologs are known for their critical role in the transport of GLUT4 and the anchoring of endocytic vesicles to the actin cytoskeleton in adipocytes (Guilherme et al, 2004a; Guilherme et al, 2004b). Ehbp1 is also recognized for its ability to promote endosomal tubulation by linking the membrane lipid PI(4,5)P2 to the actin cytoskeleton (Farmer et al, 2021; Gao et al, 2020; Wang et al, 2016), which facilitates lipid droplet engulfment during lipophagy and extending lifespan in *C. elegans* (Daniele et al, 2020; Li et al, 2016). Mechanistically, Ehbp1 acts as an effector molecule for Rab8 family members, including Rab8 and Rab10 (Li et al, 2016; Rai et al, 2016; Shi et al, 2010; Wang et al, 2016). In *Drosophila*, Ehbp1 is involved in the transport of various cargo proteins through interactions with actin, including Delta, Scabrous and Na(+)K(+) ATPase (Giagtzoglou et al, 2013; Giagtzoglou et al, 2012; Nakamura et al, 2020), highlighting its versatility in cargo transport.

In this study, we demonstrate that Ehbp1 orchestrates the polarized intracellular transport of Wg/Wnt proteins in an orderly and precise manner. The Adaptor Protein complex 1 (AP-1) interacts directly with the Wg-specific cargo adaptor Wntless (Wls), facilitating the transport of both Wg and Wls to the basolateral membrane. Ehbp1 regulates this process by competing with AP-1 for binding to Wls, leading to apical Wg transport. Thus, Ehbp1 serves as a dedicated switch that determines the distribution of Wg between the apical and basolateral cell surfaces. Using genetic and biochemical approaches, we provide mechanistic insights into this process, identifying a critical role of the coiled-coil motif within the bMERB domain of Ehbp1 in inhibiting AP-1.

Importantly, vertebrate EHBP1 also plays a similar role in regulating the polarized transport of WNT1 and WNT7A in MDCK cells. Taken together, our study identifies a conserved role for Ehbp1 in the regulation of the polarized and orderly trafficking of Wg/Wnt proteins, thus providing further insight into the fundamental mechanisms underlying Wg/Wnt signaling.

# Results

## Unraveling the role of *Ehbp1* in intracellular Wg transport

During the development of adult wings from the wing imaginal disc (Fig. 1A), Wg protein emanates from cells at the dorsal-ventral boundary and diffuses across both the basolateral and apical cell surface, activating downstream signaling (Fig. EV1A). To identify the key factors involved in the transport of Wg to both basolateral and apical cellular compartments (Fig. 1B), we conducted a comprehensive genetic screen in the developing *Drosophila* wing. Using a UAS-miRNA library (http://flybase.org/) for overexpression, we leveraged the regulatory potential of a single miRNA on multiple target genes. We then assessed the activation of Wg signaling in both adult wing tissue and larval wing imaginal discs.

Our screening strategy employed two distinct drivers: *wg-Gal4* and *hedgehog* (*hh*)-*Gal4* (Fig. 1C). The *wg-Gal4*, which is primarily active in Wg-producing cells, provided insights into Wg production and secretion, as evidenced by the functional outcomes observed in the adult wing margin phenotype. Concurrently, *hh-Gal4* facilitated the investigation of intracellular Wg trafficking within Wg-producing cells in larval wing imaginal discs, with the anterior compartment serving as a wild-type control for comparative analysis. This approach enabled us to identify a specific miRNA, *mir-bft*, and its potential target gene, *Ehbp1*, as key players in the intracellular transport of Wg from the apical to the basolateral domain.

We found that overexpression of *mir-bft* resulted in stereo-typical phenotypes indicative of decreased Wg secretion, such as the loss of sensory bristles in the adult wing blade (Fig. 1D,E) and

the intracellular accumulation of Wg protein in larval imaginal discs (Fig. 1G–G",J). The suppression of *Ehbp1*, a gene predicted to be a target of *mir-bft* (Fig. EV2A), through RNAi, yielded similar defects (Fig. 1F,H1–I). By using an endogenously tagged Wg-GFP (Port et al, 2014), we confirmed that the accumulation of intracellular Wg occurred when *Ehbp1* was suppressed by RNAi (Appendix Fig. S1). It is noteworthy that these changes in the distribution of Wg protein did not affect *wg* gene transcription, as demonstrated by *wg*-lacZ, a *wg* transcription reporter (Fig. 1J–J"). In addition, we substantiated the direct interaction between *mir-bft* and *Ehbp1*, establishing *Ehbp1* as a bona fide target of *mir-bft*. Overexpressing *mir-bft* led to a reduction in the amount of Ehbp1 protein (Fig. EV2B–B"), and a decrease in the expression of a GFP reporter linked to the *Ehbp1* 3′UTR (Fig. EV2C–C"), while no such effect was observed with a GFP reporter containing a mutated *mir-bft* binding site within the *Ehbp1* 3′UTR (Fig. EV2D–D").

To confirm the alterations in Wg signaling due to *Ehbp1* dysfunction, we examined the expression of two established Wg signaling targets, *sens* and *Dll*, in wing imaginal discs (Fig. 2A–B"). Our observations revealed an evident loss of Sens expression (Fig. 2B–B"), while Dll expression remained unaffected in somatic clones with *Ehbp1* loss-of-function (*Ehbp1$^{A28}$*) (Fig. 2A–A"). Considering that both apically and basolaterally secreted Wg proteins are required to activate short-range *sens* (Chaudhary and Boutros, 2019; Hemalatha et al, 2016), and the long-range activation of *Dll* primarily depends on Dally-like-mediated basolateral Wg spreading (Liu et al, 2022; Zecca et al, 1996), these distinct effects suggested a defect in short-range Wg spreading, implying a specific impairment in apical, but not basolateral, Wg secretion. To directly visualize Wg secretion, we employed an immunostaining method to illustrate the distribution of extracellular Wg (ExWg) on the apical and basolateral cell surfaces (Strigini and Cohen, 2000). Unlike conventional intracellular staining, this technique uses specific antibodies without permeabilizing the cells, enabling the selective detection of Wg protein in the extracellular space. This approach allows us to visualize the distribution and localization of Wg, thereby shedding light on its secretion dynamics. Indeed, when *Ehbp1* was suppressed by RNAi or in *Ehbp1* loss-of-function mutant clones, we detected an accumulation of ExWg at the basolateral surface of dorsal-ventral boundary cells (Figs. 2C1–D and 3B–B"). These findings provide strong evidence that Ehbp1 functions as an essential regulator to maintain the balance between apical and basolateral intracellular Wg transport.

## Lack of *Ehbp1* regulation on Notch, Hedgehog, and Decapentaplegic signaling in wing development

Ehbp1, known for its role in modulating the trafficking of the Notch signaling ligand Delta and the regulator Scabrous during neuron development (Giagtzoglou et al, 2013; Giagtzoglou et al, 2012), led us to explore its potential role in Notch signaling during wing development. However, our investigation revealed no significant changes in the Notch signaling pathway upon the disruption of *Ehbp1* function. Specifically, the loss of *Ehbp1* did not lead to noticeable changes in Delta or the transcriptional target Cut in *Ehbp1* loss-of-function somatic clones (Fig. EV3A–A"',B"). In addition, the functional impairment of Ehbp1 did not affect the signaling activation of the other two morphogens, Decapentaplegic

(Dpp) and Hh, which are essential for proper wing development. This was evidenced by the unaltered phosphorylation levels of the Dpp signaling activator Mothers against dpp (Mad/Smad) (Fig. EV3B"'), the Hh signaling activator Smoothened (Smo) (Fig. EV3C–C"), or the activation of the Hh signaling transcriptional factor Cubitus interruptus (CiFL) (Fig. EV3C"'). Furthermore, the distribution and levels of the morphogens Dpp and Hh remained unchanged in the presence of Ehbp1 dysfunction (Fig. EV3D–E'). The above results collectively indicate that *Ehbp1* primarily regulate Wg trafficking during wing epithelium development through a specialized mechanism dedicated to Wg transport.

## *Ehbp1* coordinates polarized Wg transport through *wls*

Given that Wntless (Wls) is recognized as an exclusive cargo receptor for Wg (Banziger et al, 2006; Bartscherer et al, 2006; Goodman et al, 2006), it is plausible to suggest that the dedicated function of Ehbp1 in Wg signaling might depend on Wls activity. To investigate this possibility, we developed a highly specific antibody against Wls, which we validated for its specificity in cells where *wls* had been selectively depleted through RNAi (Appendix Fig. S2). This antibody enabled the examination of Wls localization in Wg-producing cells (Appendix Fig. S2).

Our observations indicated that Wls was not exclusively confined to the apical compartment of the dorsal-ventral boundary cells (Appendix Fig. S2A1–A1"), as previously reported, but also exhibited a robust presence in the basolateral region (Appendix Fig. S2A2–A2"), a feature that had not been extensively studied. The apical localization of Wls is consistent with the typical location of apically secreted vesicles containing Wg proteins, which are destined for Golgi or ER recycling for subsequent rounds of Wg transport (Fig. 1B) (Belenkaya et al, 2008; Franch-Marro et al, 2008; Glaeser et al, 2018; Harterink et al, 2011; Port et al, 2008; Sun et al, 2017; Zhang et al, 2011). However, the previously less noticed presence of basolaterally localized Wls proteins may have functional implications for the direct transport of Wg to the basolateral cell surface, concurrently with the transcytosis route.

To test this hypothesis, we employed two distinct strategies to disrupt transcytosis and enhance Wls-dependent basolateral transport of Wg. First, we used RNAi to inhibit the apical secretion of Wg by suppressing specific components of the exocyst complex, *sec5* and *sec6*, which are known for their essential roles in apical Wg secretion (Chaudhary and Boutros, 2019). This approach led to a significant increase in the transport of Wls and Wg to the basolateral cellular domain (Fig. EV4A1–A2",C–F). The observed accumulation of Wg in the basolateral region is consistent with previous findings, which reported enhanced basolateral secretion of Wg following loss-of-function mutation in *sec6* or RNAi-mediated suppression of *sec6* or *sec3*, although the localization of Wls was not examined in that study (Chaudhary and Boutros, 2019). Second, we found that inhibiting transcytosis through the expression of a dominant-negative version of *gzl* (*gzl.LD*) also promoted the transport of Wls and Wg to the basolateral domain (Fig. EV4B1–B3",C–F). Consistent with our findings, previous research has noted increased co-localization of Wg and Wls upon RNAi-mediated suppression of *Klp98A*, which disrupts transcytosis, although the specific membrane domains of this Wls-Wg co-localization were not clearly defined (Witte et al, 2020). Overall, the accumulating evidence supports our conclusion that Wls proteins

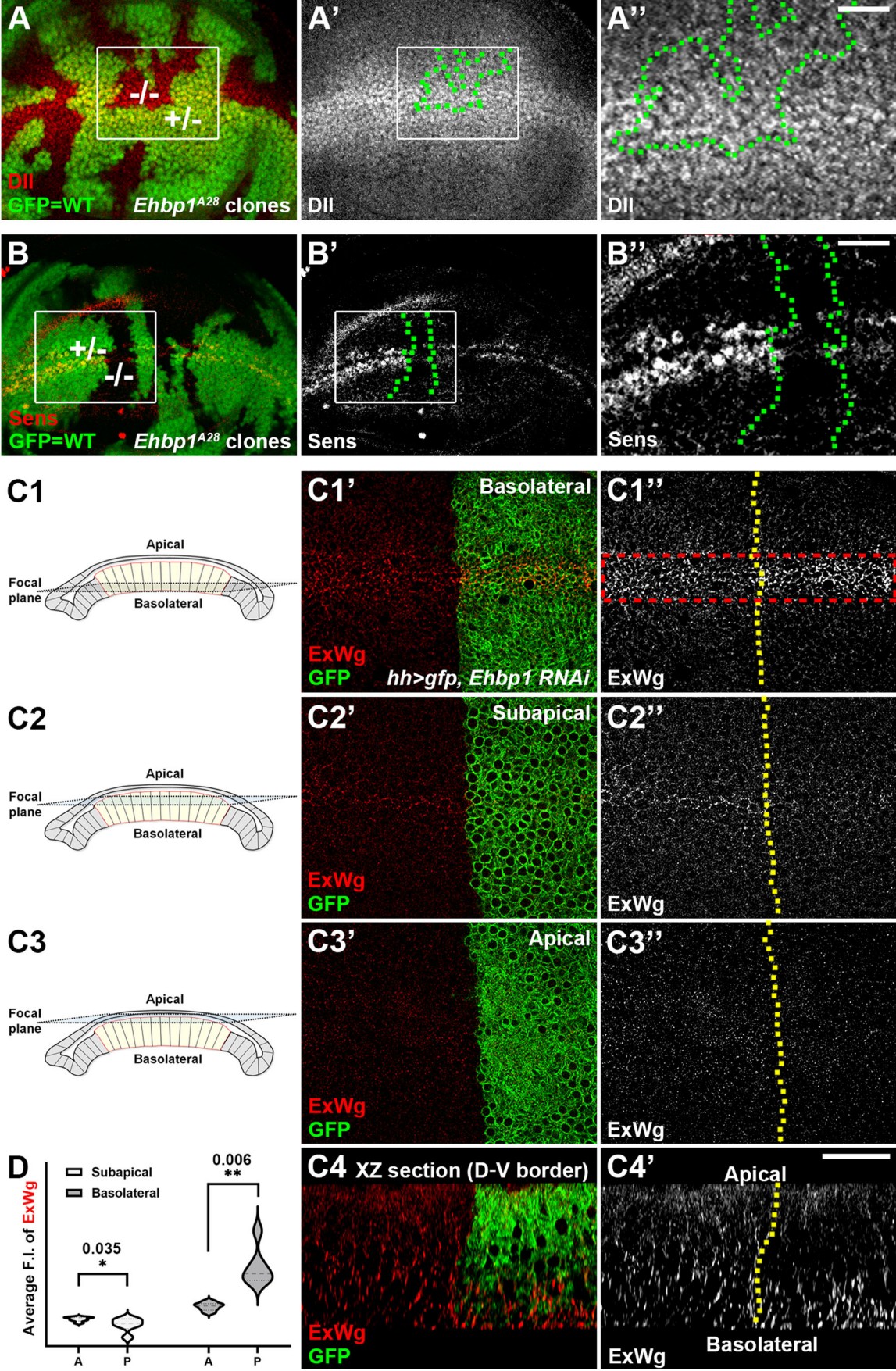

**Figure 2.  Ehbp1 maintains the proper activity of basolateral Wg transport.**

(A–B″) In the *Ehbp1^{A28}* loss-of-function mutant clones, which are negatively marked by the absence of GFP and delineated by dotted green lines, the expression of Wg signaling target Sens was down-regulated (B–B″), while Dll expression remained unaltered (A–A″). The rectangular areas marked by white lines in (A) and (A′) are enlarged in (A″), and the areas in (B) and (B′) are similarly enlarged in (B″). −/−, homozygous mutant clone cells. +/−, heterozygous cells. (C1–C4′) The basolateral (C1–C1″), subapical (C2–C2″), and apical (C3–C4″) membrane domains of immunofluorescence staining of ExWg for the indicated genotypes are shown. When RNAi against *Ehbp1* was expressed using the *hh-Gal4* driver, ExWg accumulated at the D-V boundary of basolateral domains but was reduced at the D-V boundary of the apical and subapical domains in the posterior compartments of the wing discs. A 3D reconstruction of the D-V border cells (as viewed in an XZ section) of a wing disc expressing RNAi against *Ehbp1* shows the accumulation of ExWg at the basolateral compartment (C4–C4′). Dotted yellow lines indicate the A-P boundaries. (D) Statistical analysis of ExWg immunofluorescence intensity (F.I.) at the D-V boundary in both the basolateral (C1″) and subapical sections (C2″) was performed (for each genotype, $n \geq 3$ biological replicates). Data are presented as violin plots. Two-tailed Student's t-tests were employed to analyze the differences between anterior and posterior F.I. *$p < 0.05$. **$p < 0.01$. Scale bars, 25 μm. Source data are available online for this figure.

can directly transport Wg to the basolateral cell surface, making it the primary target for Ehbp1 regulation.

In agreement with the hypothesis that *Ehbp1* regulates Wg trafficking through Wls, we found that in *Ehbp1* mutant somatic clones, Wls was retained at the basal side along with Wg (Fig. 3A1–A3‴). In addition, simultaneous RNAi-mediated suppression of *wls* and *Ehbp1* resulted in a decrease in ExWg accumulation at the basolateral domain of dorsal-ventral boundary cells (Fig. 3D,E), an effect that mimicked the outcome observed with *wls* RNAi alone (Fig. 3C–C″). Collectively, these results establish that *Ehbp1* functions through *wls* to prevent the uncontrolled basolateral transport of Wg.

## Ehbp1 antagonizes the AP-1 complex to control intracellular Wg transport

To unravel the molecular mechanisms through which *Ehbp1* regulates *wls*, we examined the interaction dynamics between Ehbp1 and Wls in cultured *Drosophila* Schneider 2 (S2) cells. Unexpectedly, we found no evidence of binding affinity between endogenous Ehbp1 and Wls (Fig. 3F). However, both proteins did interact with endogenous AP-1γ (Fig. 3F), a conserved subunit of the AP-1 complex (Nakatsu et al, 2014) with established functions in the trans-Golgi network and secretory granules (Burgess et al, 2011). Studies in polarized vertebrate Madin-Darby canine kidney (MDCK) cells have suggested that AP-1 activity is necessary for the basal secretion of WNT proteins (Yamamoto et al, 2013; Yamamoto et al, 2015; Yamamoto et al, 2017). This observation raises a possibility that the basolateral transport of Wls may hinge on the functionality of the AP-1 complex, a process that could be precisely monitored in the presence of Ehbp1.

To validate our hypothesis, we investigated the localization of Wls in somatic clones with disrupted AP-1 complex function. Loss-of-function mutations in *AP-1γ* or *AP-1μ* resulted in a noticeable decrease in basolateral Wls localization, particularly along the dorsal-ventral boundary of wing discs (Fig. 4A–B′). This reduction in basolateral Wls distribution correlated with diminished ExWg at the basolateral cell surface and an increase at the apical side in *AP-1γ* mutant somatic clones (Fig. 4C1–C2′). As a result, the generation of *AP-1γ* or *AP-1μ* mutant clones in the wing discs led to stereotypical phenotypes in the wing blades, indicating reduced Wg signaling, such as the loss of marginal tissues and sensory bristles in the adult *Drosophila* wing blade (Fig. 4D,E). Furthermore, the co-expression of *AP-1γ* RNAi and *Ehbp1* RNAi resulted in the enrichment of ExWg at the apical cell surface (Fig. 4G1–G2′,H,I), resembling the phenotype observed with *AP-1γ*

RNAi alone (Fig. 4F1–F2′,H,I). These results support the central role of Ehbp1 in orchestrating the basolateral and apical transport of Wls and Wg, actively counteracting the AP-1 complex in Wg-producing cells. In alignment with this model, overexpression of *Ehbp1* reduced the amount of Wls protein co-immunoprecipitated with AP-1γ in S2 cells (Fig. 5B). Consequently, a higher concentration of ExWg was detected at the apical cell surface, rather than the basal side, in wing imaginal discs when *Ehbp1* was overexpressed (Figs. 5A1–A2' and 4H,I).

## The coiled-coil motifs are required for Ehbp1 to antagonize the AP-1 complex

The Ehbp1 protein consists of three main domains: the N-terminal C2 (Nt-C2) domain, the central calponin homology (CH) domain, and the C-terminal bivalent Mical/EHBP Rab binding (bMERB) domain (Fig. 5C). Overexpression of Ehbp1 mutants lacking the Nt-C2 (Ehbp1ΔNt-C2) or CH (Ehbp1ΔCH) domain, but not the bMERB (Ehbp1ΔbMERB) domain, retained the ability to reduce the association between Wls protein and AP-1γ in S2 cells (Fig. 5D). Strikingly, the sole removal of the two Coiled-coil motifs within the bMERB domain (Ehbp1ΔCC) alone was sufficient to disrupt the inhibitory effect of Ehbp1 on the interaction between Wls and AP-1 (Fig. 5D). This disruption was further substantiated by the failure of Ehbp1ΔbMERB and Ehbp1ΔCC to rescue the accumulation of Wls and Wg at basolateral cellular domains in *Ehbp1* mutant clones (Fig. 5F–G‴), in contrast to the effective rescue achieved by wild-type Ehbp1, Ehbp1ΔNt-C2, and Ehbp1ΔCH (Fig. 5E–E‴; Appendix Fig. S3A–B‴).

To underscore the essential role of the Coiled-coil motifs in regulating Wls and Wg transport, we generated *Ehbp1* mutant flies specifically engineered to lack these Coiled-coil motifs (Appendix Fig. S3C). Indeed, somatic clones of this mutant displayed an evident enrichment of Wls and Wg at basolateral cellular domains (Fig. 5H–H‴). These cumulative findings highlight the critical contribution of the Coiled-coil motifs in maintaining the proper activity of basolateral transport of Wls and Wg, emphasizing their crucial role in the regulatory mechanism governed by Ehbp1 in this cellular process.

## Neither Rab8 nor Rab10 is essential for the intracellular transport of Wg

Previous studies have suggested that Ehbp1 serves as an effector molecule for members of the Rab8 family, including Rab8 and Rab10 (Li et al, 2016; Rai et al, 2016; Shi et al, 2010; Wang et al, 2016). To investigate the potential roles of Rab8 and Rab10 in the

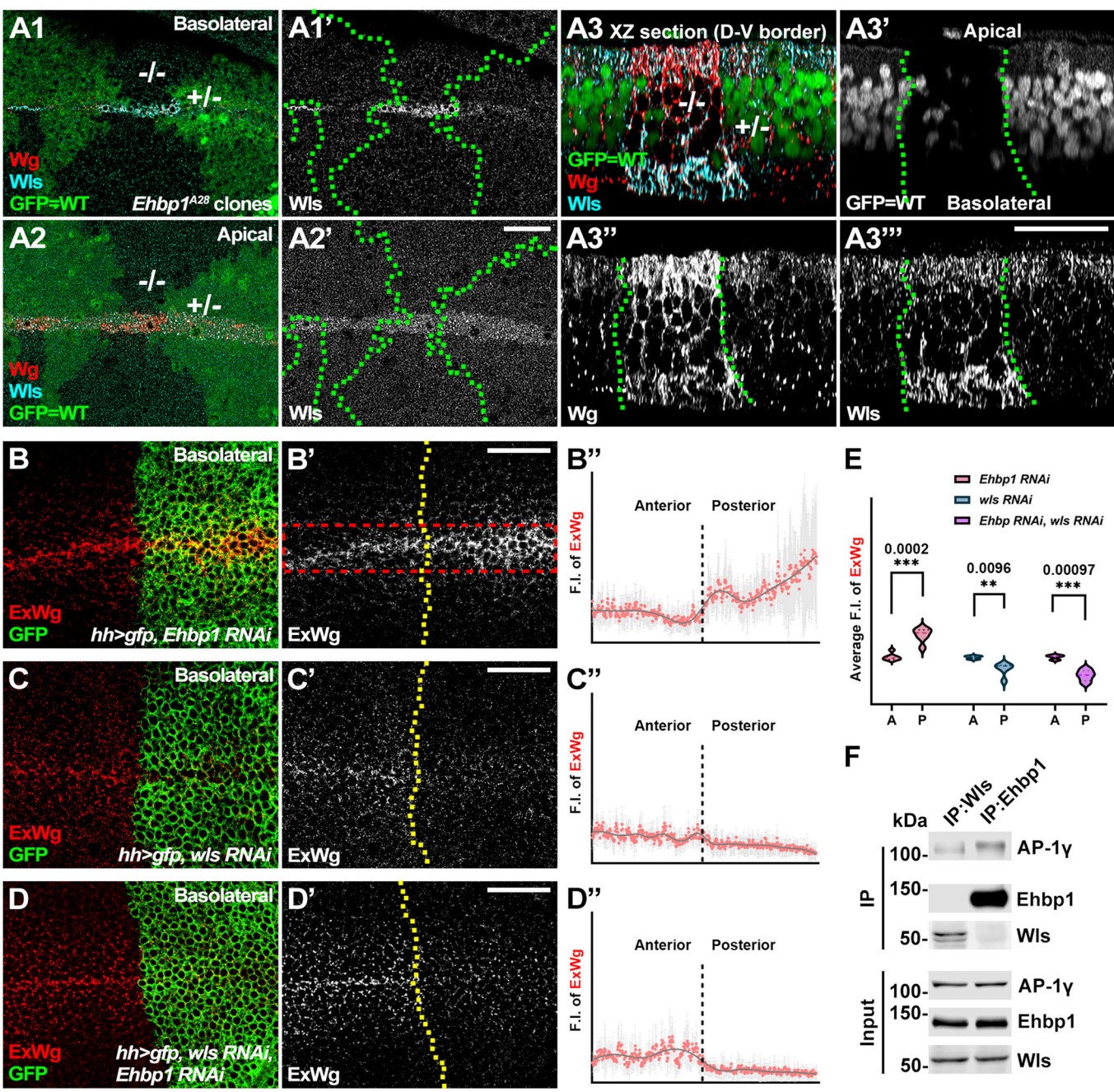

**Figure 3. *Ehbp1* functions through *wls* to regulate polarized Wg transport.**

(**A1–A3'''**) In negatively marked *Ehbp1^A28* mutant somatic clones (marked by the absence of GFP, outlined by dotted green lines), Wls accumulated at the D-V boundary in the basolateral domains of the wing discs (**A1–A1'**). A 3D reconstruction of the D-V border cells (as viewed in an XZ section) of the wing disc bearing negatively marked *Ehbp1^A28* mutant clones shows the accumulation of Wls and Wg at the basolateral domain (**A3–A3'''**). −/−, homozygous mutant clone cells. +/−, heterozygous cells. (**B–B''**) Upon expression of RNAi against *Ehbp1* using the *hh-Gal4* driver, ExWg accumulated at the D-V boundary in the basolateral domains of the posterior compartment of the wing disc. Dotted yellow lines indicate the A-P boundaries. The plot profiles of immunofluorescence staining were generated within the rectangular areas demarcated by red dashed lines (for each genotype, n ≥ 3 biological replicates). Data are shown as mean ± SD. (**C–D''**) When RNAi against *wls* was expressed alone (**C–C''**), or in combination with *Ehbp1* (**D–D''**), using the *hh-Gal4* driver, ExWg was reduced at the D-V boundary in the basolateral domains of the posterior compartments of the wing discs. Dotted yellow lines indicate the A-P boundaries. Plot profiles of immunofluorescence staining in (**C'**) and (**D'**) were generated (**C''** and **D''**) (for each genotype, n ≥ 3 biological replicates). Data are shown as mean ± SD. (**E**) Statistical analysis of ExWg immunofluorescence intensity (F.I.) in (**D''**), (**E''**), and (**F''**) was performed (for each genotype, n ≥ 3 biological replicates). Data are presented as violin plots. Two-tailed Student's t-tests were employed to analyze the differences between anterior and posterior F.I. **p < 0.01. ***p < 0.001. (**F**) Endogenous Ehbp1 and Wls were co-immunoprecipitated with endogenous AP-1γ in S2 cell lysates. However, endogenous Ehbp1 and Wls did not immunoprecipitate with each other. Scale bars, 25 μm. Source data are available online for this figure.

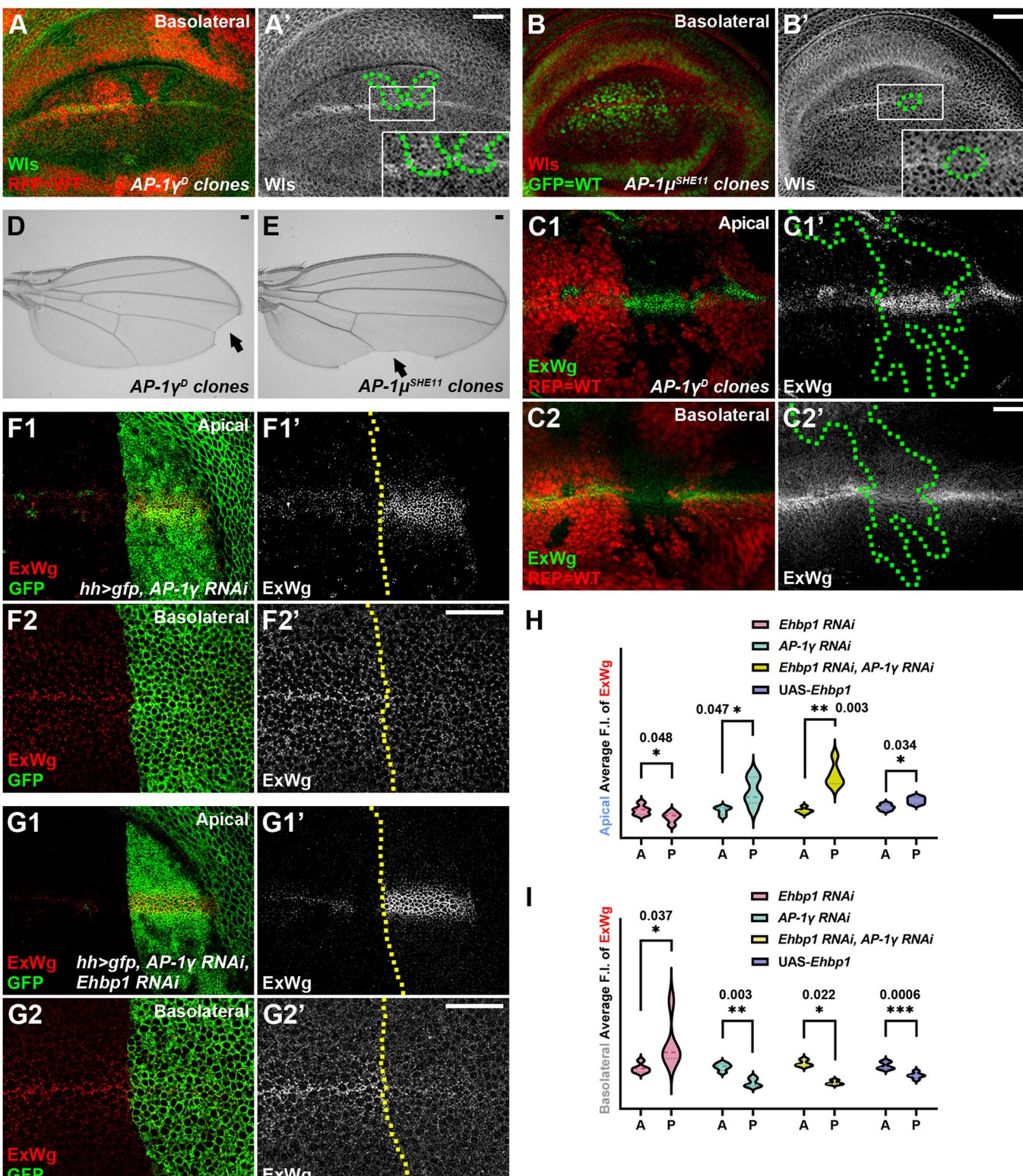

regulation of intracellular Wg transport by Ehbp1, we examined the localization of Wg under conditions of enhanced or altered Rab8 or Rab10 function (Zhang et al, 2007). Surprisingly, the overexpression of constitutively active (CA) or dominant negative (DN) forms of *Rab8* or *Rab10* did not appear to alter the intracellular Wg

transport (Fig. EV5A–A",C–C",E–E",G–G"), suggesting that Rab8 and Rab10 may not be essential for the intracellular transport of Wg under normal physiological conditions.

However, when *Ehbp1* RNAi was co-expressed with *Rab8CA/ DN* or *Rab10CA/DN*, we observed intracellular accumulation of Wg

◄

**Figure 4. Ehbp1 antagonizes the AP-1 complex to regulate polarized Wg transport.**

(A–B') In the negatively marked $AP-1\gamma^D$ (A–A') and $AP-1\mu^{SHE11}$ (B–B') loss-of-function mutant somatic clones (marked by the absence of GFP, outlined by dotted green lines), Wls was reduced at the D-V boundary in the basolateral domains of the wing discs. The rectangular areas marked by white lines in (A') and (B") are shown as magnified insets. (C1–C2') In the negatively marked $AP-1\gamma^D$ mutant somatic clones (marked by the absence of GFP, outlined by dotted green lines), ExWg at the D-V boundary was reduced in the basolateral domains (C2–C2'), while it accumulated in the apical domains of the wing disc (C1–C1'). (D, E) Loss of the Wg signaling-induced wing margin phenotype in adult wing blades associated with $AP-1\gamma^D$ or $AP-1\mu^{SHE11}$ mutant somatic clones. Arrows indicate the presence of a serrated wing margin. (F1–G2') RNAi-mediated knockdown of $AP-1\gamma$ (F1–F2') alone or in combination with $Ehbp1$ (G1–G2') using the $hh$-Gal4 driver led to the accumulation of ExWg at the D-V boundary in the apical domains, while it was reduced in the basolateral domains of the posterior compartments of the wing discs. Dotted yellow lines indicate the A-P boundaries. (H, I) Statistical analysis was performed on the ExWg immunofluorescence intensity (F.I.) at the D-V boundary, comparing both the basolateral and apical membrane domains in (F1–G2') and Fig. 5A1–A2' (for each genotype, $n \geq 3$ biological replicates). Data are presented as violin plots. Two-tailed Student's t-tests were used to analyze the differences between anterior and posterior F.I. $*p < 0.05$. $**p < 0.01$. $***p < 0.001$. Scale bars, 25 μm. Source data are available online for this figure.

protein (Fig. EV5B–B",D–D",F–F",H–H"). This intracellular accumulation generally resembled that seen with $Ehbp1$ RNAi alone, it is noteworthy that the co-expression of $Ehbp1$ RNAi and $Rab10CA$ led to a significant apical accumulation of Wg protein (Fig. EV5F–F"), suggesting a potential synergistic role for Rab10CA and Ehbp1 in intracellular Wg distribution. In addition, $Ehbp1$ RNAi notably altered the localization of Rab8CA or Rab10CA from the apical to basolateral compartment (Fig. EV5B–B',F–F'), indicating a potential feedback influence of Enbp1 on the intracellular trafficking of activated Rab proteins. In summary, while Rab8 family proteins may not be required for normal intracellular Wg transport, their influence becomes apparent and may synergize with Ehbp1 only under conditions where Ehbp1 function is compromised. Further investigation is required to fully appreciate the underlying mechanisms and the significance of these observations.

## Conserved regulation of intracellular WNT transport by Ehbp1 in vertebrate cells

Consistent with the role of AP-1 in facilitating the basolateral transport of Wls during Wg trafficking in *Drosophila* wing development, a comparable association has been observed in vertebrates between the controlled transport of WLS and the release of WNT proteins (Yamamoto et al, 2013; Yamamoto et al, 2015; Yamamoto et al, 2017). This correlation suggests that the vertebrate counterpart of Ehbp1 may play a similar role in coordinating the transport of Wnt proteins to both the apical and basolateral domains.

To investigate this hypothesis, we focused on WNT1, a key member of the WNT family. In MDCK epithelial cells, which are known for their well-defined apical-basal polarity, overexpressed WNT1 has been observed to be secreted through both apical and basolateral pathways, a process facilitated by WLS (Yamamoto et al, 2017). To delve deeper into this process, we established a stable MDCK cell line expressing WNT1-GFP (MDCK/WNT1) and conducted experiments to determine the impact of *EHBP1* suppression on the secretion of WNT1-GFP. These MDCK/WNT1 cells were cultured as a monolayer on a filter support in a transwell system (Yamamoto et al, 2017). To trace cell-surface-associated WNT1, the apical and basolateral membranes of the cells were separately labeled with sulfo-NHS-LC-biotin. This labeling allowed for the isolation of biotinylated WNT1-GFP proteins from the cell surface using Streptavidin magnetic beads (Fig. 6A). RNAi-mediated suppression of *EHBP1* specifically increased the presence of WNT1-GFP at the basolateral membrane (Fig. 6A), leading to enhanced basolateral secretion in the polarized MDCK/WNT1 cells.

Having established that EHBP1 regulates the orderly transport of overexpressed WNT proteins in MDCK cells, we next sought to determine whether endogenous WNT proteins are also subject to EHBP1 regulation. As *WNT1* is not expressed in MDCK cells, we focused on the endogenously expressed WNT family genes for which we had available antibodies for immunofluorescence staining. We found that WNT7A, a WNT family protein highly expressed in MDCK cells (Yamamoto et al, 2015), is primarily localized to the basolateral compartment (Fig. 6B) and exhibits significant basolateral secretion, as confirmed by extracellular WNT7A (ExWNT7A) staining (Fig. 6C). Reduction of *EHBP1* expression using siRNA led to a notable accumulation of WNT7A proteins in the basolateral cell compartment and an increase in basolateral secretion (Fig. 6D,E). Knockdown of *WLS* resulted in significant intracellular retention of WNT7A, effectively preventing its basolateral secretion (Fig. 6F,G). When both *WLS* and *EHBP1* were suppressed simultaneously (Fig. 6H,I), the resulting phenotypes were similar to those observed with *WLS* RNAi alone (Fig. 6F,G). These findings are consistent with our observations in the *Drosophila* wing imaginal disc.

Given the critical role of AP-1 in the basolateral transport of WNT proteins, and the fact that WNT7A is primarily secreted basolaterally, it was anticipated that knocking down *AP-1 μ1A* would lead to intracellular enrichment of WNT7A and a reduction in basolateral secretion (Fig. 6J,K). However, in contrast to our observations in *Drosophila*, we did not detect increased apical ExWNT7A staining, which could be due to less effective staining on the apical cell surface or a lack of intracellular apical transport machinery for WNT7A in MDCK cells. Consistent with Ehbp1's role in competing with WLS for AP-1 binding to prevent unregulated basolateral Wg transport in *Drosophila*, simultaneous suppression of *AP-1 μ1A* and *EHBP1* yielded phenotypes resembling those seen with *AP-1 μ1A* RNAi alone (Fig. 6L,M). Collectively, these findings indicate that EHBP1, WLS, and AP-1 are crucial for the proper trafficking and localization of WNT proteins in MDCK cells, providing strong evidence for the conservation of the role of Ehbp1 in the polarized secretion of Wnt proteins from flies to vertebrates.

## Discussion

Protein trafficking is a precisely regulated process that is crucial for the proper functioning of key signaling pathways, particularly the Wg/Wnt signaling pathway. Despite extensive research on the extracellular trafficking of Wg (Gross et al, 2012; Han et al, 2005; Huang and Kornberg, 2015; McGough et al, 2020; Port and Basler,

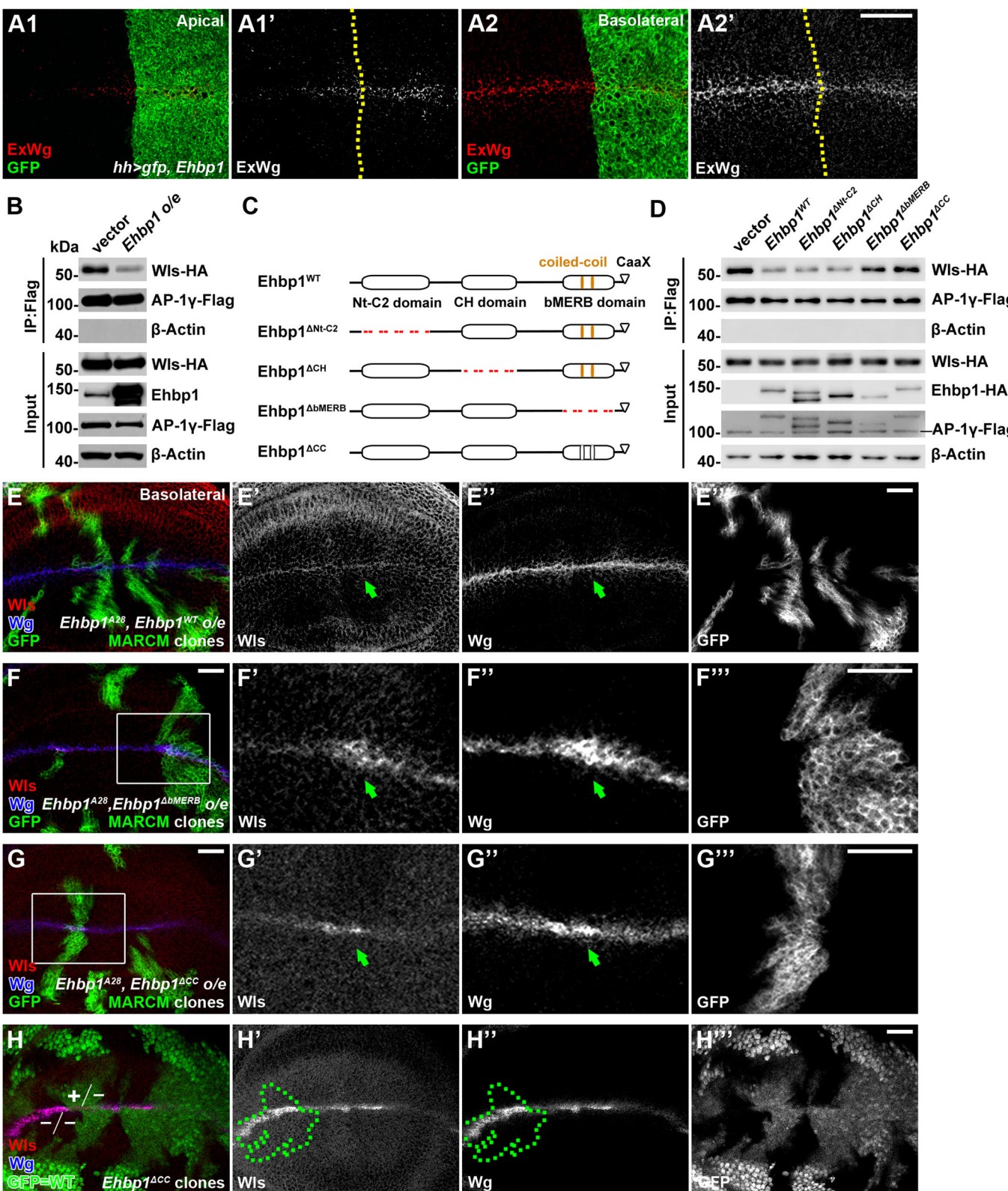

2010; Routledge and Scholpp, 2019) and Wnt morphogens (Greco et al, 2001; Gross et al, 2012; Hsieh et al, 1999; Kakugawa et al, 2015; Langton et al, 2016; Leyns et al, 1997; McGough et al, 2020; Mihara et al, 2016), the mechanisms governing their intracellular transport within polarized Wg/Wnt-producing cells have remained elusive. Our study aims to decipher these complexities, with a specific focus on the pivotal role played by Ehbp1 in the trafficking of Wg/Wnt family proteins.

◄ **Figure 5. The Coiled-coil motifs are essential for Ehbp1 to antagonize AP-1 complex and orchestrate Wls and Wg intracellular transport in wing discs.**

(A1–A2′) When *Ehbp1* was overexpressed using the *hh-Gal4* driver, ExWg accumulated at the D-V boundary in the apical domain (**A** and **A′**), whereas its levels were reduced in the basolateral domain of the posterior compartment of the wing disc (**B** and **B′**). Dotted yellow lines indicate the A-P boundaries. (**B**) Overexpression of *Ehbp1* reduced the amount of Wls protein that co-immunoprecipitated with AP-1γ in S2 cell lysates. (**C**) Schematic showing the wild-type and various mutant forms of Ehbp1. Wild-type Ehbp1 contains an N-terminal C2 (Nt-C2) domain, a calponin homology (CH) domain, and a C-terminal bivalent Mical/EHBP Rab binding (bMERB) domain. Two coiled-coil motifs are localized within the bMERB domain. (**D**) Overexpression of *Ehbp1^WT^*, *Ehbp1^ΔNt-C2^*, or *Ehbp1^ΔCH^* was capable of reducing the amount of Wls co-immunoprecipitated with AP-1γ in S2 cell lysates, while overexpressing *Ehbp1^ΔbMERB^* or *Ehbp1^ΔCC^* had no such effect. (**E–E‴**) Expression of *Ehbp1^WT^* in *Ehbp1^A28^* mutant somatic clones (positively marked by GFP) fully rescued the accumulation of Wg and Wls at the D-V boundary in the basolateral domain of the wing disc. Arrows indicate the somatic clones at the D-V boundary. (**F–G‴**) Expression of *Ehbp1^ΔbMERB^* or *Ehbp1^ΔCC^* in *Ehbp1^A28^* mutant somatic clones (positively marked by GFP) failed to rescue the accumulation of Wg and Wls at the D-V boundary in the basolateral domain of the wing disc. The rectangular areas marked by white lines in (**F**) and (**G**) are shown in magnified (**F′–F‴**) and (**G′–G‴**). Arrows indicate the somatic clones at the D-V boundary. (**H–H″**) In negatively marked *Ehbp1^ΔCC^* mutant somatic clones (marked by the absence of GFP, outlined by dotted green lines), Wg and Wls accumulated at the D-V boundary in the basolateral domain of the wing disc, resembling the defects observed in *Ehbp1^A28^* loss-of-function mutant clones (c.f. panels 4A1–A3‴). Arrows indicate the somatic clones at the D-V boundary. −/−, homozygous mutant clone cells. +/−, heterozygous cells. Scale bars, 25 µm. Source data are available online for this figure.

The initial concept of apical-to-basal transcytosis as the primary mechanism for Wg intracellular transport (Gallet et al, 2008; Witte et al, 2020; Yamazaki et al, 2016) has faced challenges. Debates have emerged regarding the functional significance of Wg transcytosis, considering its minimal impact on Wg signaling even when transcytosis is reduced or disrupted (Witte et al, 2020). Our study reveals the complexities of Wg/Wnt protein trafficking, providing compelling evidence for an alternative pathway that directly guides Wg/Wnt proteins to the basolateral surface. This alternative route is orchestrated by a trio of proteins: Ehbp1, AP-1, and Wls (Fig. 7). Disruption of Ehbp1 or AP-1 function results in the accumulation of Wg on specific cell surfaces, leading to phenotypes stereo-typically associated with reduced activation of Wg signaling. This newly identified pathway aligns with the established role of AP-1 and Wls in the basal secretion of Wnt family proteins in mammalian cells, including WNT1, WNT3A, and WNT5A (Yamamoto et al, 2013; Yamamoto et al, 2015; Yamamoto et al, 2017), as well as WNT7A in our study. However, the involvement of AP-1 and Ehbp1 in the apical secretion of other Wnt family members, such as WNT11A, requires further investigation. Since AP-1 acts as a general adaptor that facilitates polarized sorting in the trans-Golgi network and/or in the recycling endosomes, our findings suggest that Ehbp1 serves as a key determinant, specifying the sorting of Wg by AP-1, highlighting the coordination of apical and basolateral trafficking routes for Wg in this newly uncovered pathway.

In addition to its essential role in regulating intracellular Wg trafficking, Ehbp1 in *Drosophila* has been shown to be involved in the transport of various cargo proteins, including Delta, Scabrous and Na(+)K(+)ATPase (Giagtzoglou et al, 2013; Giagtzoglou et al, 2012; Nakamura et al, 2020), suggesting that the transport mechanisms of Scabrous and Na(+)K(+)ATPase may also rely on the antagonistic interaction between the Ehbp1 and AP-1 complex, as identified in our study. Our findings are consistent with the observation that in the *Ehbp1* mutant, Scabrous accumulates with the AP-1 complex, thereby impeding its secretion in fly photoreceptors (Giagtzoglou et al, 2013). However, in contrast to our results, disruption of the function of either the Ehbp1 or AP-1 complex has been found to lead to the mislocalization of the basolateral protein Na(+)K(+)ATPase to the apical stalk cell surface in fly photoreceptors (Nakamura et al, 2020). This cooperative role of Ehbp1 and AP-1 in regulating basal protein transport contrasts sharply with the antagonistic

relationship observed between Ehbp1 and AP-1 in the coordinated polarized trafficking of Wg proteins. Nevertheless, the known interaction of Ehbp1 with the exocyst subunit Sec15, demonstrated in *Drosophila* S2 Cells (Giagtzoglou et al, 2012), may provide potential mechanistic insights into how Ehbp1 coordinates the apical secretion of Wg through the exocyst complex.

Our findings, coupled with evidence of Wg signaling occurring at apical surfaces and within endosomal compartments (Chaudhary and Boutros, 2019; Hemalatha et al, 2016; Linnemannstons et al, 2020; Marois et al, 2006), highlight the significance of both basolateral and apical transport of Wg. Interestingly, studies in MDCK cells have revealed distinctions between apically secreted hydrophilic Wnt proteins and their basally secreted lipophilic counterparts (Yamamoto et al, 2013; Yamamoto et al, 2015; Yamamoto et al, 2017). To address the bidirectional nature of Wg transport and the coordination of proteins with differing hydro-philic or hydrophobic properties, our study suggests a potential role for Ehbp1 in this process. The bidirectional trafficking, reflected by the selective sorting of hydrophilic and hydrophobic proteins within the Wg-Wls complex, introduces additional complexities to this vital cellular process. Consequently, our study implies that Wg signaling activation should be considered in conjunction with the equilibrium of bidirectional Wg transport, rather than as a result of unidirectional transport towards either the apical or basolateral side. Considering the lipid modifications observed in several developmental signaling molecules such as Hh/Sonic Hh (Shh) and Dpp/TGF-β, do these morphogens also have specific cargo sorting mechanisms similar to Ehbp1? Under-standing whether Hh/Shh and Dpp/TGF-β rely on dedicated intracellular transport machinery for their sorting and trafficking processes, similar to Wg/Wnt, could provide valuable insights into the broader understanding of morphogen transport in develop-mental processes.

On a mechanistic level, human EHBP1 functions as an effector molecule for members of the Rab8 family, including Rab8 and Rab10 (Rai et al, 2016). In the absence of Rab8 family proteins, the bMERB domain of human EHBP1 forms an intramolecular complex with the CH domain, leading to auto-inhibition and preventing actin binding. The binding of Rab8 to the bMERB domain alleviates this inhibition (Rai et al, 2020). This mechanism is supported by observations that Ehbp1 depletion results in endocytic recycling defects in the intestine and nonpolarized germline cells, which resemble phenotypes observed in *C. elegans*

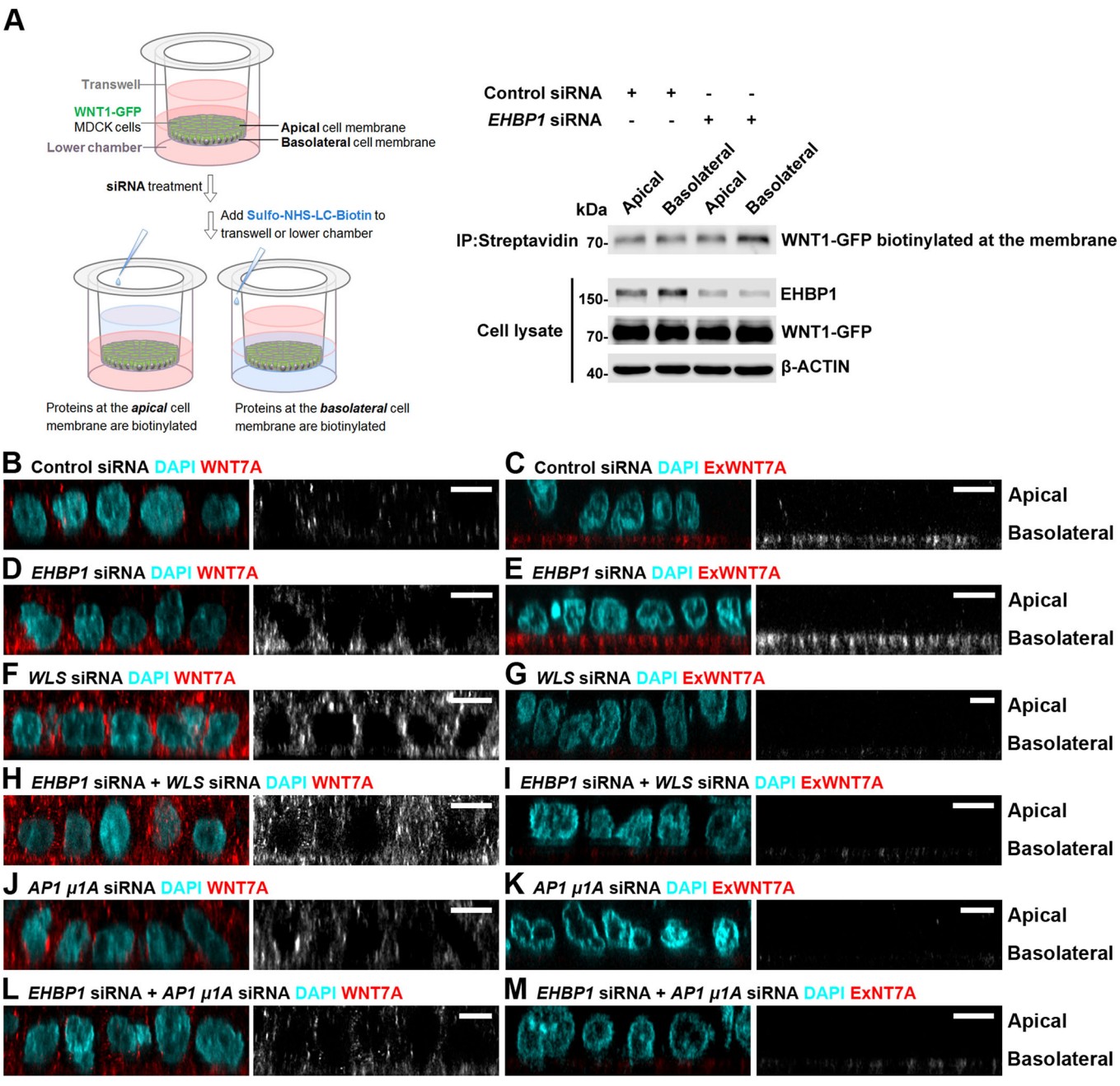

Figure 6. The regulation of intracellular WNT1 transport by Ehbp1 is conserved in vertebrate cells.

(A) Schematic showing the transwell assay (left). MDCK cells were grown as a monolayer on a filter support in a transwell system (left). Sulfo-NHS-LC-biotin was used to selectively label the apical and basolateral cell membranes, and the resulting biotinylated cell-surface-associated WNT1-GFP proteins were isolated using Streptavidin magnetic beads. Knocking down *EHBP1* using siRNA selectively increased the presence of WNT1-GFP at the basolateral membrane (right). (B–M) 3D reconstructions of MDCK monolayer cells are shown. In wild-type MDCK cells, WNT7A predominantly localized to the basolateral compartment (B), with ExWNT7A primarily detected at the basolateral cell surface (C). In MDCK cells expressing siRNA against *EHBP1*, WNT7A accumulated significantly in the basolateral compartment (D), and ExWNT7A accumulated at the basolateral cell surface (E). In MDCK cells expressing siRNA against *WLS*, WNT7A accumulated throughout the cell (F), with reduced ExWNT7A at the basolateral surface (G). When siRNA against *EHBP1* and *WLS* were co-expressed in MDCK cells, WNT7A proteins were significantly enriched throughout the cell (H), and ExWNT7A were reduced at the basolateral cell surface (I). MDCK cells expressing siRNA against *AP-1 μ1A* showed WNT7A accumulation within the cell (J), with a decrease of ExWNT7A at the basolateral surface (K). Co-expression of siRNA against *EHBP1* and *AP-1 μ1A* in MDCK cells resulted in WNT7A proteins accumulating within the cell (L), and a decrease in ExWNT7A at the basolateral cell surface (M). Scale bars, 10 μm. Source data are available online for this figure.

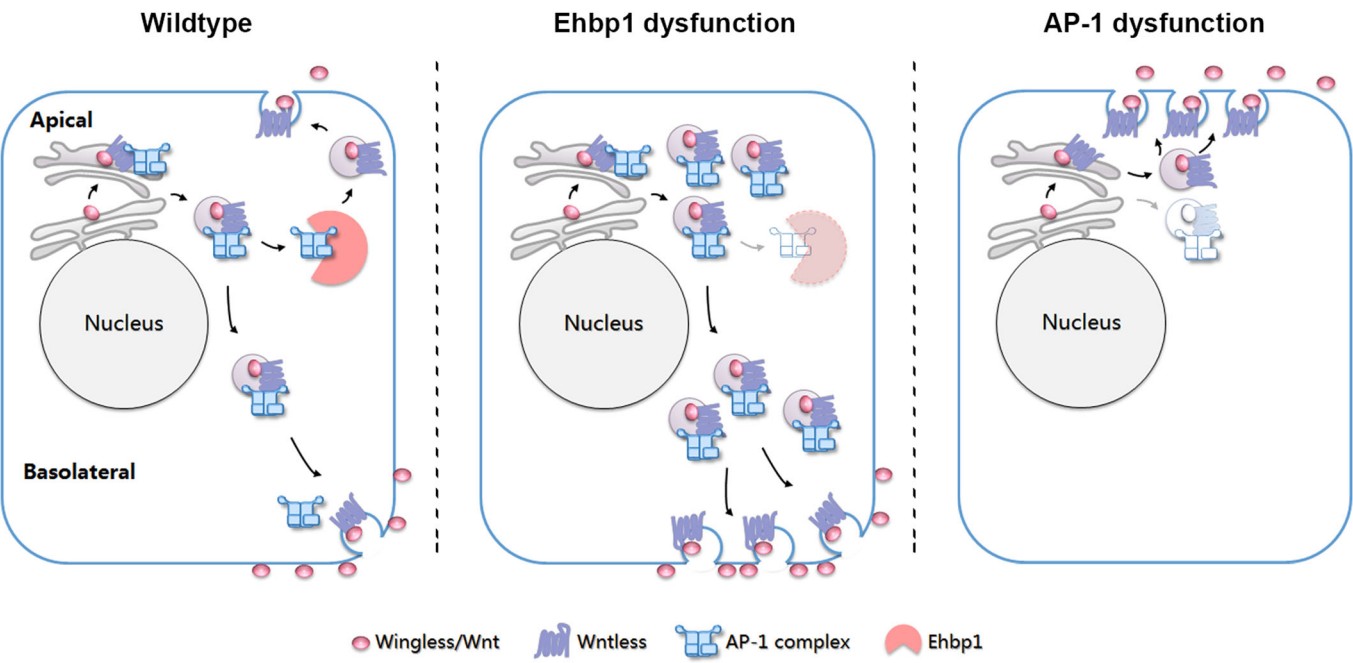

**Figure 7. Regulation of polarized intracellular Wg/Wnt transport by Ehbp1, AP-1, and Wls.**

The schematic illustrates the mechanism controlling the intracellular transport of Wg/Wnt proteins, which involves three ley components: Ehbp1, AP-1, and Wls. Upon synthesis in the trans-Golgi network (TGN), Wg/Wnt proteins are associated with Wls and AP-1 complex. By default, vesicles containing Wg/Wnt-Wls are targeted for transport to the basolateral cell surface. Ehbp1 interacts with the AP-1 complex, competing with the Wls, and this interaction diverts Wg/Wnt-Wls from the default basolateral route. Consequently, in *Ehbp1*-dysfunctional cells, the basolateral transport of Wg/Wnt-Wls is significantly enhanced. Conversely, if AP-1 is defective, the majority of Wg/Wnt-Wls complexes are directed to the apical cell surface. Since both AP-1 and *Ehbp1* dysfunction results in signaling defects, a balanced polarized distribution of Wg/Wnt is crucial for maintaining accurate signaling outcomes.

with Rab8/10 deletions (Shi et al, 2010). In addition, the binding of Rab10 to Enbp1 enhances its function in promoting endosomal tubulation (Wang et al, 2016). During lipophagy, the activation of Rab10 increases its association with mammalian EHBP1, facilitating the engulfment of lipid droplets in hepatocytes (Li et al, 2016). Contrary to previous findings, our study found that neither Rab8 nor Rab10 is required for the intracellular transport of Wg in *Drosophila*. This contradicts earlier evidence suggesting the involvement of Rab8a vesicles in the regulation of Wnt ligand delivery in the intestinal stem cell niche (Das et al, 2015) and the role of Rab8 in AP-1B-dependent basolateral transport in MDCK cells (Ang et al, 2003), as well as EHBP1L1-Bin1-dynamin dependent apical-directed transport in small intestine organoids (Nakajo et al, 2016). While Rab10 has been reported to facilitate the selective transport of basement membrane proteins to the basal cell surface in *Drosophila* follicle cells (Isabella and Horne-Badovinac, 2016; Lerner et al, 2013), our findings suggest that other Rab family members, such as Rab11 (Giagtzoglou et al, 2012), may play a role in regulating the basolateral transport of Wg upstream of Ehbp1. This hypothesis warrants further investigation. Moreover, the mechanistic insights into the function of Ehbp1 presented in this study may reveal an alternative pathway for Wg transport that has not been previously characterized, offering new perspectives on the underlying biological processes.

## Methods

### Reagents and tools table

| Reagent/Resource | Reference or Source | Identifier or Catalog Number |
|---|---|---|
| **Experimental Models** | | |
| MDCK (*C. familiaris*) | ATCC | CRL-2935, CVCL_B033 |
| Schneider 2 (*D. melanogaster*) | ATCC | CRL-1963, CVCL_Z232 |
| *Drosophila, dpp-gfp/TM3* | A gift of Tom Kornberg | N/A |
| *Drosophila, FRT82B AP-1μ[SHE11]/TM6B* | A gift of Roland Le Borgne | N/A |
| *Drosophila, hh-gfp* (II) | A gift of Tom Kornberg | N/A |
| *Drosophila, hs-Flp[122]; FRT42D, M2(58)f, ubi-gfp* | A gift of Jessica Treisman | N/A |
| *Drosophila, hs-Flp[122]; FRT82B, ubi-gfp* | A gift of Jessica Treisman | N/A |
| *Drosophila, UAS-gzl-LD* (II) | A gift of Jean-Paul Vincent | N/A |
| *Drosophila, w ; FRT42D, Ehbp1[A28]/CyO, hs-hid* | A gift of Hugo Bellen | N/A |
| *Drosophila, wg-gfp* (II) | A gift of Fillip Port | N/A |
| *Drosophila, ap-Gal4/Cyo* | Bloomington *Drosophila* Stock Center | BDSC Cat# 3041 |

| Reagent/Resource | Reference or Source | Identifier or Catalog Number |
|---|---|---|
| Drosophila, UAS-bft (III) | Bloomington Drosophila Stock Center | BDSC Cat# 41133 |
| Drosophila, UAS-gzl-RNAi (II) | Bloomington Drosophila Stock Center | BDSC Cat# 64034 |
| Drosophila, wg-Gal4 (II) | Bloomington Drosophila Stock Center | BDSC Cat# 4918 |
| Drosophila, whi-mRFP(nls), hs-Flp,FRT19A | Bloomington Drosophila Stock Center | BDSC Cat# 31418 |
| Drosophila, wg-lacZ/CyO | Bloomington Drosophila Stock Center | BDSC Cat# 11205 |
| Drosophila, yw, AP-1γ[D], FRT19A/FM7C, Kr-Gal4, UAS-gfp, sn+ | Bloomington Drosophila Stock Center | BDSC Cat# 57052 |
| Drosophila, UAS-gzl-RNAi (II) | National Institute of Genetics | NIG Cat# 10277R-3 |
| Drosophila, UAS-Ehbp1 RNAi (II) | TsingHua Fly Center | THFC Cat# THU02340.N |
| Drosophila, UAS-sec5 RNAi (III) | TsingHua Fly Center | THFC Cat# THU2688 |
| Drosophila, UAS-sec6 RNAi (III) | TsingHua Fly Center | THFC Cat# THU2636 |
| Drosophila, UAS-w RNAi (III) | TsingHua Fly Center | THFC Cat# THU0583 |
| Drosophila, UAS-AP-1γ RNAi/TM6B | Vienna Drosophila RNAi Center | VDRC Cat# 3275 |
| Drosophila, UAS-wls RNAi (III) | Vienna Drosophila RNAi Center | VDRC Cat# 5214 |
| Drosophila, ap-Gal4, UAS-myr-mRFP (II) | This paper | N/A |
| Drosophila, ap-Gal4, UAS-myr-mRFP, hh-gfp (II) | This paper | N/A |
| Drosophila, hh-Gal4, UAS-mCD8-gfp/TM6B | This paper | N/A |
| Drosophila, hh-Gal4, UAS-myr-mRFP/TM6B | This paper | N/A |
| Drosophila, hs-Flp[122], tub-Gal4, UAS-gfp; FRT42D, tubulin-Gal80 | This paper | N/A |
| Drosophila, tub:gfp:3'UTR[Ehbp1] (II) | This paper | N/A |
| Drosophila, tub:gfp:3'UTRmut[Ehbp1] (II) | This paper | N/A |
| Drosophila, tub-Gal80[ts]; hh-Gal4, UAS-mCD8-gfp/TM6B | This paper | N/A |
| Drosophila, UAS-Ehbp1[wt] (III) | This paper | N/A |
| Drosophila, UAS-Ehbp1[ΔNt-C2] (III) | This paper | N/A |
| Drosophila, UAS-Ehbp1[ΔCH] (III) | This paper | N/A |
| Drosophila, UAS-Ehbp1[ΔbMERB] (III) | This paper | N/A |
| Drosophila, UAS-Ehbp1[ΔCC] (III) | This paper | N/A |
| Drosophila, w ; FRT42D, Ehbp1[ΔCC]/CyO | This paper | N/A |
| **Recombinant DNA** | | |
| Plasmid: pSIN-egfp | A gift of Jianguo Chen | N/A |
| Plasmid: pCFD4 | Addgene | Cat No. #83954 |
| Plasmid: pMD2.G | Addgene | Cat No. #12259 |
| Plasmid: psPAX2 | Addgene | Cat No. #12260 |
| Plasmid: pActin-Electra2 | Addgene | Cat No. #184942 |
| Plasmid: pActin5.1-bgal-V5His | Addgene | Cat No. #136238 |
| Plasmid: pActin5.1-3×flag-AP-1γ | This paper | N/A |
| Plasmid: pCaSpeR | DGRC | Cat No. #1187 |

| Reagent/Resource | Reference or Source | Identifier or Catalog Number |
|---|---|---|
| Plasmid: pCaSpeR-hs | DGRC | Cat No. #1215 |
| Plasmid: pCaSpeR-hs-wls-HA | This paper | N/A |
| Plasmid: pCaSpeR-tub:gfp:3'UTR[Ehbp1] | This paper | N/A |
| Plasmid: pCaSpeR-tub:gfp:3'UTRmut[Ehbp1] | This paper | N/A |
| Plasmid: pCFD4-gRNAdCC1-gRNAdCC2 | This paper | N/A |
| Plasmid: pGEMT-3P2A | Addgene | Cat No. #111772 |
| Plasmid: pGEMT-Ehbp1-dCC1-dCC2 | This paper | N/A |
| Plasmid: pSIN-WNT1-egfp | This paper | N/A |
| Plasmid: pUAST-Gr64f | Addgene | Cat No. #21080 |
| Plasmid: pUAST-Ehbp1[wt]-2×HA | This paper | N/A |
| Plasmid: pUAST-Ehbp1[ΔNt-C2]-2×HA | This paper | N/A |
| Plasmid: pUAST-Ehbp1[ΔCH]-2×HA | This paper | N/A |
| Plasmid: pUAST-Ehbp1[ΔbMERB]-2×HA | This paper | N/A |
| Plasmid: pUAST-Ehbp1[ΔCC]-2×HA | This paper | N/A |
| **Antibodies** | | |
| Mouse anti-Dll | A gift of Ian Duncan | N/A |
| Guinea pig anti-Sens | A gift of Hugo Bellen | N/A |
| Rabbit anti-β-Actin | ABclonal Technology | Cat# AC026, AB_2768234 |
| Mouse anti-HA (6E2) | Cell signaling Technology | Cat# 2367, AB_10691311 |
| Rabbit anti-pSmad (D5B10) | Cell signaling Technology | Cat# 13820, AB_2493181 |
| Rat anti-Ci full-length (2A1) | Developmental Studies Hybridoma Bank | AB_2109711 |
| Mouse anti-Cut (2B10) | Developmental Studies Hybridoma Bank | AB_528186 |
| Mouse anti-β-Galactosidase (40-1a) | Developmental Studies Hybridoma Bank | AB_528100 |
| Mouse anti-Delta (C594.9B) | Developmental Studies Hybridoma Bank | AB_528194 |
| Mouse anti-Smo (20C6) | Developmental Studies Hybridoma Bank | AB_528472 |
| Mouse anti-Wg (4D4) | Developmental Studies Hybridoma Bank | AB_528512 |
| Mouse anti-FLAG | Engibody Biotechnolog | Cat# AT0022 |
| Rabbit anti-EHBP1 | Proteintech | Cat# 17637-1-AP, AB_2097216 |
| Rabbit anti-Wnt7A | Proteintech | Cat# 27177-1-AP, AB_3085932 |
| Rabbit anti-GFP | Thermo Fisher Scientific | Cat# A-11122, AB_221569 |
| Rabbit anti-AP-1γ | This paper | N/A |
| Rabbit anti-Ehbp1 | This paper | N/A |
| Rabbit anti-Wls | This paper | N/A |
| Rat anti-Wls | This paper | N/A |
| Goat anti-Mouse IgG (H + L), Alexa Fluor™ 488 | Thermo Fisher Scientific | Cat# A-11001 |

| Reagent/Resource | Reference or Source | Identifier or Catalog Number |
|---|---|---|
| Goat anti-Mouse IgG (H + L), Alexa Fluor™ 568 | Thermo Fisher Scientific | Cat# A-11004 |
| Goat anti-Rabbit IgG (H + L), Alexa Fluor™ 488 | Thermo Fisher Scientific | Cat# A-11008 |
| Goat anti-Rabbit IgG (H + L), Alexa Fluor™ 568 | Thermo Fisher Scientific | Cat# A-11011 |
| Goat anti-Rat IgG (H + L), Alexa Fluor™ 488 | Thermo Fisher Scientific | Cat# A-11006 |
| Goat anti-Rat IgG (H + L), Alexa Fluor™ 568 | Thermo Fisher Scientific | Cat# A-11077 |
| Goat anti-Rat IgG (H + L), Alexa Fluor™ 647 | Thermo Fisher Scientific | Cat# A-21247 |
| Goat anti-Mouse IgG (H + L), Alexa Fluor™ 405 | Thermo Fisher Scientific | Cat# A-31553 |
| Goat anti-Guinea pig IgG (H + L), Alexa Fluor™ 568 | Thermo Fisher Scientific | Cat# A-11075 |
| HRP conjugated Goat anti-Mouse | Abclonal | Cat# AS003 |
| HRP conjugated Goat anti-Rabbit | Abclonal | Cat# AS014 |
| HRP conjugated Goat anti-Rat | Abclonal | Cat# AS028 |
| **Oligonucleotides and other sequence-based reagents** | | |
| PCR primers | This study | Appendix Table S2 |
| **Chemicals, Enzymes and other reagents** | | |
| IPTG | AMRESCO | Cat# 0487 |
| Fetal bovine serum | Lanzhou Bailing | Cat# 20130507 |
| Magzol | Magen | Cat# R4801-02 |
| Protease inhibitor cocktail | Roche | Cat# 13538100 |
| 3x FLAG peptide | Sigma-Aldrich | Cat# F4799 |
| NP-40 (IGEPAL®CA-630) | Sigma-Aldrich | Cat# I8896 |
| Poly-L-Lysine | Sigma-Aldrich | Cat# P4707 |
| Puromycin | Sigma-Aldrich | Cat# 540411 |
| Triton X-100 | Sigma-Aldrich | Cat# T8787 |
| Trizma base (Tris) | Sigma-Aldrich | Cat# T1503 |
| Tween-20 | Sigma-Aldrich | Cat# P1379 |
| DMEM medium | Thermo Fisher Scientific | Cat# C11995500BT |
| Penicillin-Streptomycin | Thermo Fisher Scientific | Cat# 15140122 |
| Schneider's Drosophila medium | Thermo Fisher Scientific | Cat# 21720024 |
| Sulfo-NHS-LC-Biotin | APExBIO | Cat# A8003 |
| Euparal mounting medium | BioQuip | Cat# 6372A |
| GFP-Nanoab-Magnetic Beads | LABLEAD | Cat# GNM-50-2000 |
| Streptavidin Magnetic Beads | MedChemExpress | Cat# HY-K0208 |
| Eastep RT Master Mix Kit | Promega | Cat# LS2050 |
| RiboMAX™ Large Scale RNA Production System | Promega | Cat# P1280 |
| Agarose anti-FLAG | Sigma-Aldrich | Cat# A2220 |
| Dynabeads™ Protein A Immunoprecipitation Kit | Thermo Fisher Scientific | Cat# 10006D |
| Lipofectamine™ 2000 Transfection Reagent | Thermo Fisher Scientific | Cat# 11668019 |
| Oligo(dT)$_{12-18}$ Primer | Thermo Fisher Scientific | Cat# 18418012 |
| SuperScript™ III reverse transcriptase | Thermo Fisher Scientific | Cat# 18080093 |
| KOD-Plus-Neo | TOYOBO | Cat# KOD-401 |

| Reagent/Resource | Reference or Source | Identifier or Catalog Number |
|---|---|---|
| VECTASHIELD® Antifade Mounting Medium | Vector Laboratories | Cat# H-1200 |
| Trans T1 | TransGen Biotech | Cat# CD501 |
| **Software** | | |
| Photoshop CS 5 | Adobe | https://www.adobe.com |
| GraphPad Prism 8 | GraphPad | http://www.graphpad.com |
| Leica LAS X | Leica Microsystems | https://www.leica-microsystems.com |
| Image J 1.53 | National Institutes of Health | https://imagej.nih.gov |
| QCapture Pro 5 | QImaging Corporation | https://www.qimaging.com |
| **Other** | | |
| Leica DM IL LED microscope | Leica Microsystems | https://www.leica-microsystems.com |
| Leica TCS SP8 inverted confocal microscope | Leica Microsystems | https://www.leica-microsystems.com |
| 4×/0.10 objective lense | Leica Microsystems | https://www.leica-microsystems.com |
| 10×/0.22 objective lense | Leica Microsystems | https://www.leica-microsystems.com |
| 20×/0.70 objective lense | Leica Microsystems | https://www.leica-microsystems.com |
| 63×/1.40 oil objective lense | Leica Microsystems | https://www.leica-microsystems.com |
| QImaging MicroPublisher RTV-5.0 CCD Camera | QImaging Corporation | https://www.photometrics.com |

## Fly genetics

Standard procedures were followed for fly cultures and crosses. *Drosophila* stocks were either ordered from stock centers or obtained from other laboratories, and they are listed as follows: *FRT42D, Ehbp1$^{A28}$/CyO, hs-hid* (a gift of Hugo Bellen, Baylor College of Medicine, Houston, TX), *hh-gfp* (II) and *dpp-gfp/TM3* (a gift of Tom Kornberg, University of California, San Francisco, CA), *FRT82B AP-1μ$^{[SHE11]}$/TM6B* (a gift of Roland Le Borgne, Université de Rennes, Rennes Cedex, France), *hs-Flp$^{122}$; FRT42D, M2(58)f, ubi-gfp* (a gift of Jessica Treisman, New York University, New York, NY), *UAS-gzl-LD* (II) (a gift of Jean-Paul Vincent, The Francis Crick Institute, London, UK), *wg-gfp* (II) (a gift of Fillip Port, Heidelberg University, Heidelberg, Germany), *ap-Gal4/CyO* (Bloomington Stock Center, BDSC Cat# 3041; RRID:BDSC_3041), *UAS-bft* (III) (BDSC; Cat# 41133; RRID:BDSC_41133), *UAS-gzl-RNAi* (II) (BDSC; Cat# 64034; RRID:BDSC_64034), *UAS-yfp-Rab8DN.[T22N]* (BDSC; Cat# 9780; RRID:BDSC_9780), *UAS-yfp-Rab8CA.[Q67L]* (BDSC; Cat# 9781; RRID:BDSC_9781), *UAS-yfp-Rab10DN.[T23N]* (BDSC; Cat# 9786; RRID:BDSC_9786), *UAS-yfp-Rab10CA.[Q68L]* (BDSC; Cat# 9787; RRID:BDSC_9787), *wg-Gal4* (II) (BDSC; Cat# 4918; RRID:BDSC_4918), *whi-mRFP(nls), hs-Flp, FRT19A* (BDSC Cat# 31418; RRID:BDSC_31418), *wg-lacZ/CyO* (BDSC Cat# 11205; RRID:BDSC_11205), *yw, AP-1γ[D], FRT19A/FM7C, Kr-Gal4, UAS-gfp, sn+* (BDSC Cat# 57052; RRID:BDSC_57052), *UAS-gzl-RNAi* (II) (National Institute of Genetics, NIG; Cat# 10277R-3), *UAS-Ehbp1 RNAi* (II) (TsingHua Fly Center, THFC; Cat# THU02340.N), *UAS-sec5 RNAi* (III)

(THFC; Cat# THU2688), *UAS-sec6 RNAi* (III) (THFC; Cat# THU2636), *UAS-w RNAi* (III) (THFC; Cat# THU0583), *UAS-AP-1γ RNAi/TM6B* (Vienna RNAi Center, VDRC; Cat# 3275), *UAS-wls RNAi* (III), (VDRC; Cat# 5214). Detailed genetic crosses corresponding to each figure are provided in Appendix Table S1.

Wild-type and domain-deletion *UAS-Ehbp1* transgenic flies, as well as *tub:gfp:3′UTR^Ehbp1^* and *tub:gfp:3′UTRmut^Ehbp1^* sensor lines, were generated through φC31 transgenic integration. The *Ehbp1^△CC^* flies were generated using CRISPR-Cas9-mediated homology-directed repair (HDR) (Port et al, 2014). Two plasmids were injected into the eggs of *nos-Cas9* transgenic flies to express two gRNAs and provide the HDR templates. The resulting F1 offspring were collected and crossed with individual chromosome balancer flies. PCR identification was used to confirm the deletion of the two coiled-coil motifs. The identified *Ehbp1^△CC^* allele was subsequently recombined with *FRT42D* for further studies.

Details regarding the primers used in fly transformation constructs and genotyping identification can be found in Appendix Table S2.

## Immunofluorescence staining of wing imaginal discs and adult wing imaging

Immunofluorescence staining was conducted following established protocols. Briefly, wing discs dissected from third-instar larvae were fixed in 4% paraformaldehyde, then blocked in a solution of PBS containing 0.1% Triton-X100 and 0.2% BSA for 1 h. Subsequently, they were incubated overnight at 4 °C with the following primary antibodies: Mouse anti-β-Galactosidase (1:100; 40-1a; Developmental Studies Hybridoma Bank, DSHB; RRID:AB_2109711), Rat anti-Ci full-length (1:10; 2A1; DSHB; RRID:AB_2109711), Mouse anti-Cut (1:200; 2B10; DSHB; RRID:AB_528186), Mouse anti-Delta (1:100; C594.9B; DSHB; RRID:AB_528194), Mouse anti-Dll (1:200; a gift from Ian Duncan, University of California, Berkeley, CA), Rabbit anti-Ehbp1 (1:2000; generated in this study), Guinea pig Rabbit anti-pSmad (1:500; D5B10; Cell Signaling; RRID:AB_2493181), anti-Sens (1:2000; a gift from Hugo Bellen), Mouse anti-Smo (1:20; 20C6; DSHB; RRID:AB_528472), Mouse anti-Wg (1:200; 4D4; DSHB; RRID:AB_528512), Rat anti-Wls (1:5000; generated in this study). Following the primary antibody incubation, the wing discs were washed in PBS containing 0.1% Triton-X100 for 5 min, 3–5 times, and then incubated with Alexa Fluor-conjugated secondary antibodies (1:400; Thermo Fisher; Cat# A-11001, A-11004, A-11006, A-11008, A-11011, A-11075, A-11077, A-21247, and A-31553) for 1 h at room temperature, followed by another 3–5 washes before mounting in VECTASHIELD (Vector Laboratories; Cat# H-1200). For Immunofluorescence staining in MDCK cells, Triton-X100 was omitted from all solutions, with the exception of a 5-min treatment involving PBS containing 0.1% Triton-X100 after fixation. The primary antibody employed was rabbit anti-hEHBP1 (1:100; Proteintech; Cat# 17637-1-AP; RRID:AB_2097216). Images were captured using a Leica TCS SP8 confocal microscope equipped with Leica 20×/0.70 and 63×/1.40 oil objective lenses, and the image data were processed using LAS AF X (Leica Microsystems) software and Adobe Photoshop CS5 (RRID:SCR_014199).

For extracellular Wg (ExWg) staining, third-instar larvae were dissected in ice-cold S2 cell culture medium on a cold coverslip placed on a block of blue ice. The wing discs were then incubated in ice-cold S2 medium with mouse anti-Wg antibody (1:10; 4D4; DSHB; RRID:AB_528512) in an Eppendorf tube, entirely submerged in ice, for 30–60 min. After incubation, the discs were washed with ice-cold PBS 3–5 times, followed by fixation for 20 min in ice-cold PBS containing 4% formaldehyde (Strigini and Cohen, 2000). The subsequent steps were consistent with the conventional antibody staining protocol.

Adult wings were dissected and mounted in Euparal mounting medium (BioQuip; Cat# 6372A). The images were acquired using a Leica DMIL LED inverted microscope equipped with Leica 4×/0.10 and 10×/0.22 objective lenses and a QImaging MicroPublisher RTV-5.0 CCD Camera (QImaging).

## Molecular biology

The construction of the *tub:gfp:3′UTR^Ehbp1^* sensor vector involved the fusion of the *3′UTR^Ehbp1^* sequence after the *gfp* coding sequence. This construct was subsequently cloned into the *pCaSpeR* plasmid (DGRC; Cat# 1187; RRID:DGRC_1187), which contains the *tubulin* promoter and *AttB* site. The *3′UTR^Ehbp1^* sequence was cloned from cDNA obtained from RNA extracted from *yw* flies. The *tub:gfp:3′UTRmut^Ehbp1^* sensor vector was derived from the wild-type sensor vector by introducing a mutation in the *mir-bft* binding site (*GTTGCCAT* to *GTTGGCTT*) through site-directed mutagenesis (TOYOBO; Cat# KOD-401). Both sensor vectors were utilized for the generation of transgenic flies.

For the generation of *Ehbp1* overexpression transgenic flies, the full-length *Ehbp1* and four domain-deleted versions of *Ehbp1* were cloned into the *pUAST* vector (Addgene; Cat# 21080; RRID:Addgene_21080) as PCR fragments, cut with *NotI*. This plasmid features a C-terminal *2×HA* tag and an *AttB* site, allowing for φC31-mediated integration into *AttP* sites. All domain-deleted versions of *Ehbp1* were amplified using primers designed to span the deleted domains.

To generate *Ehbp1^△CC^* transgenic flies, two gRNAs were cloned into the *pCFD4* plasmid (Addgene; Cat# 83954; RRID:Addgene_83954) using the *gCC1-F* and *gCC2-R* primers. Three fragments of *Ehbp1* genome sequences, corresponding to regions upstream of *CC1*, between *CC1* and *CC2*, and downstream of *CC2*, were amplified using their respective primers and then assembled into the *pGEMT* plasmid (Addgene; Cat# 111772; RRID:Addgene_111772) as a template for DNA repair.

The full-length *AP-1γ* and *AP-1μ* were cloned into the *pAC5.1* vector (Addgene; Cat# 136238; RRID:Addgene_136238), which features an N-terminal 3×flag tag for cell transfection. The full-length *wls* was cloned into the *pCaSpeR-hs* vector (DGRC, Cat# 1215; RRID:DGRC_1215), derived from *pCaSpeR*, featuring a C-terminal HA tag for cell transfection. The full-length human *WNT1* was cloned into the *pSIN-egfp* vector (a gift of Jianguo Chen, Peking University, Beijing, China) to generate a cell line stably expressing *WNT1-egfp*.

## Antibody generation

A polyclonal rabbit anti-Ehbp1 antibody was developed against the full-length *Drosophila* Ehbp1 protein by ABclonal. The full-length *Drosophila* Ehbp1 cDNA was cloned into the *pUAST* vector to serve as the template for antibody generation by ABclonal. This antibody was used in immunoprecipitation, immunoblotting, and immunofluorescence analyses.

Polyclonal antibodies against full-length *Drosophila* Wls were generated by ABclonal. The full-length *Drosophila wls* cDNA was cloned into the *pCaSpeR* vector, serving as the template for ABclonal's antibody production process. This procedure resulted in the production of both a Rabbit anti-Wls and a Rat anti-Wls antibody. The Rabbit anti-Wls antibody was used in immunoprecipitation and immunoblotting, whereas the Rat anti-Wls antibody was used in immunofluorescence and immunoblotting experiments.

A polyclonal Rabbit anti-AP-1γ antibody was developed against the full-length *Drosophila* AP-1γ protein by ABclonal. The full-length *Drosophila AP-1γ* cDNA was cloned into the *pActin* vector (Addgene; Cat# 184942; RRID:Addgene_184942), which was used as the template for ABclonal's antibody production. This antibody was used in immunoblotting experiments.

The specificity of the above antibodies was confirmed through immunofluorescence and immunoblotting analyses.

## Cell culture and transfection

*Drosophila* Schneider S2 cells (ATCC; CRL-1963; RRID:CVCL_Z232) were cultured at 25 °C in Schneider's *Drosophila* medium (Thermo Fisher; Cat# 21720024), supplemented with 10% FBS (Lanzhou Bailing; Cat# 20130507) and 100 U/ml of penicillin/streptomycin (Thermo Fisher; Cat# 15140122). DNA transfection was conducted using a standard calcium phosphate protocol.

HEK293T (ATCC; CRL-1573; RRID:CVCL_0063) and MDCK cells (ATCC; CRL-2935; RRID:CVCL_0422) were cultured in DMEM medium (Thermo Fisher; Cat# C11995500BT) supplemented with 10% FBS and 100 U/ml of penicillin/streptomycin at 37 °C. siRNA oligos were generated by Tsingke Biotech. DNA and siRNA transfection was carried out using the Lipofectamine™ 2000 Transfection Reagent (Thermo Fisher; Cat# 11668019). For the infection of MDCK cells, lentiviruses were packaged in HEK293T cells. Briefly, *pSIN-WNT1-egfp*, *psPAX2* (Addgene; Cat# 12260; RRID:Addgene_12260), and *pMD2.G* (Addgene; Cat# 12259; RRID:Addgene_12259) were transfected into HEK293T cells. Conditioned media were collected 48 h post-transfection and subsequently added to MDCK cells for infection. After 48 h, 1.5 µg/mL puromycin (Sigma-Aldrich; Cat# 540411) was added to the culture media for selection.

## Transwell assay and cell immunofluorescence staining

Transwell assay was performed as previously described (Yamamoto et al, 2013). MDCK cells were seeded on transwell polycarbonate filters (LABSELECT; Cat#14111) and allowed to grow until they formed a confluent monolayer. After a polarized monolayer was formed, the intracellular distribution and secretion of Wnt proteins were examined using an apical-basolateral sorting assay and cell immunofluorescence staining.

MDCK cells expressing *WNT1-gfp* (suspended in 0.7 ml of culture medium) were seeded on transwell polycarbonate filters. Subsequently, 1.4 ml of culture medium was added to the lower chamber for a 3-day incubation. To identify membrane cell-surface-associated WNT1, the apical and basolateral surface membranes of the cells were selectively incubated with 0.5 mg/ml sulfo-NHS-LC-biotin (APExBIO) for 30 min at 4 °C. Biotinylated proteins were subsequently precipitated using Streptavidin

Magnetic Beads (MedChemExpress; Cat# HY-K0208), and the resulting precipitates were probed using a GFP antibody (1:1000; Thermo Fisher; Cat# A-11122; RRID:AB_221569).

To visualize the intracellular localization of WNT7A in MDCK cells cultured on transwell polycarbonate filters, the cells were fixed for 15 min in PBS containing 4% paraformaldehyde, followed by permeabilization with PBS containing 0.5% Triton X-100. After permeabilization, the cells were blocked for 1 h in a solution of PBS containing 0.1% Triton-X100 and 0.2% BSA, and then incubated overnight at 4 °C with a rabbit anti-WNT7A antibody (1:100; Proteintech; Cat# 27177-1-AP; RRID: AB_3085932). The cells were washed 3–5 times for 5 min each in PBS containing 0.1% Triton-X100, followed by incubation with Alexa Fluor-conjugated secondary antibodies (1:400; Thermo Fisher; A-11011) for 1 h at room temperature. After the secondary antibody incubation, an additional 3–5 washed were performed before mounting the cells in VECTASHIELD (Vector Laboratories; Cat# H-1200).

To visualize basolaterally secreted WNT7A, the cells were incubated with the anti-WNT7A antibody for 2 h at 37 °C without permeabilization. Subsequently, the cells were fixed in PBS containing 4% paraformaldehyde for 15 min at room temperature, blocked for 1 h in PBS containing 0.2% BSA, and then incubated with Alexa Fluor-conjugated secondary antibodies (1:400; Thermo Fisher; A-11011) for 1 h at room temperature. Finally, the cells were washed 3–5 times before mounted in VECTASHIELD (Vector Laboratories; Cat# H-1200).

## RNA isolation and RT-PCR

Total RNA from fly wing imaginal discs or HEK293T cells was extracted utilizing Magzol reagent (Magen; Cat# R4801-02) following standard procedures. Residual genomic DNA was removed using the gDNA remover provided in the Promega Eastep RT Master Mix Kit (Promega; Cat# LS2050). Subsequently, first-strand cDNA was synthesized using an oligo-dT primer (Thermo Fisher Scientific; Cat# 18418012) and SuperScript III reverse transcriptase (Thermo Fisher; Cat# 18080093).

## Biochemistry

S2 cells, HEK293T cells, and third-instar larval wing imaginal discs were lysed in NP-40 buffer (1% NP-40, 150 mM NaCl, 50 mM Tris-HCl, pH 8) or RIPA buffer (1% Triton X-100, 0.1% SDS, 1% Sodium deoxycholate, 150 mM NaCl, 50 mM Tris-HCl, pH 7.4), both supplemented with a protease inhibitor cocktail (Roche). Immunoblotting was conducted following standard protocols. The following antibodies were used for immunoblotting: rabbit anti-AP-1γ (1:5000, generated for this study), rabbit anti-β-Actin (1:100,000; ABclonal Cat# AC026; RRID:AB_2768234), rabbit anti-Ehbp1 (1:50,000, generated for this study), mouse anti-Flag (1:1000; Engibody Biotechnology Cat# AT0022), mouse anti-HA (1:1000; 6E2; Cell Signaling; RRID:AB_10691311), rabbit anti-Wls (1:20,000, generated for this study), rat anti-Wls (1:20,000, generated for this study), and HRP conjugated goat anti-mouse/rabbit/rat IgG (H + L; 1:10,000; ABclonal; Cat# AS003, AS014, and AS028).

Immunoprecipitation was carried out using either agarose anti-flag (Sigma-Aldrich; Cat# A2220) or the Dynabeads™ Protein A Immunoprecipitation Kit (Thermo Fisher Scientific; Cat# 10006D)

following the manufacturer's instructions. Agarose anti-flag beads were eluted using 3×flag peptide (Sigma-Aldrich; Cat# F4799) following the manufacturer's recommendations. The immunoblots presented in all figures are representative of at least three independent experiments.

## Quantification and statistical analysis

To quantify the intensity of antibody staining, images were captured with consistent confocal settings, and fluorescence intensity was assessed using NIH ImageJ (RRID:SCR_003070). Specifically, the plot profile was measured within the defined test area in ImageJ to obtain the fluorescence intensity profile. As this profile covers regions of different genotypes, the average value of each profile region was computed as the fluorescence intensity corresponding to the respective genotype. Two-tailed Student's t-tests were used to assess differences between two distinct genotypes.

## Data availability

This study includes no data deposited in external repositories.

The source data of this paper are collected in the following database record: biostudies:S-SCDT-10_1038-S44319-024-00289-1.

## Peer review information

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

## Acknowledgements

We thank Drs. Hugo Bellen, Jianguo Chen, Ian Duncan, Tom Kornberg, Roland Le Borgne, Jessica Treisman, Jean-Paul Vincent, Bloomington *Drosophila* Stock Center (BDSC) at Indiana University, Bloomington, Tsinghua Fly Center at Tsinghua University (THFC), Beijing, Vienna *Drosophila* RNAi Center (VDRC) at Vienna Biocenter Core Facilities, Vienna, and Developmental Studies

Hybridoma Bank (DSHB) at the University of Iowa, Iowa City (created by the Eunice Kennedy Shriver National Institute of Child Health and Human Development of the National Institutes of Health and maintained at the University of Iowa, Iowa City) for fly stocks, antibodies and plasmids. We also thank Dr. Chunyan Shan at the National Center for Protein Science at Peking University for assistance with microscopic imaging. This work was supported by grants from the National Natural Science Foundation of China (32170716 to M Liu, 32330026 to AJ Zhu, and 32300990 to Y Zhang), National Key Research and Development Program of China (2021YFA0805800 to AJ Zhu and M Liu), Qidong-SLS Innovation Fund (to AJ Zhu), the Peking-Tsinghua Center for Life Sciences (to AJ Zhu and M Liu) and the Ministry of Education Key Laboratory of Cell Proliferation and Differentiation (to AJ Zhu). Y Zhang was supported by a Peking University President's Scholarship and a Boya Postdoctoral Fellowship.

## Author contributions

**Yuan Gao**: Conceptualization; Formal analysis; Investigation; Writing—original draft; Writing—review and editing. **Jing Feng**: Formal analysis; Validation; Investigation; Writing—original draft; Writing—review and editing. **Yansong Zhang**: Investigation. **Mengyuan Yi**: Investigation. **Lebing Zhang**: Investigation. **Yan Yan**: Writing—original draft; Writing—review and editing. **Alan Jian Zhu**: Conceptualization; Supervision; Funding acquisition; Writing—original draft; Project administration; Writing—review and editing. **Min Liu**: Conceptualization; Supervision; Funding acquisition; Investigation; Writing—original draft; Project administration; Writing—review and editing.

Source data underlying figure panels in this paper may have individual authorship assigned. Where available, figure panel/source data authorship is listed in the following database record: biostudies:S-SCDT-10_1038-S44319-024-00289-1.

## Disclosure and competing interests statement

The authors declare no competing interests.

# Expanded View Figures

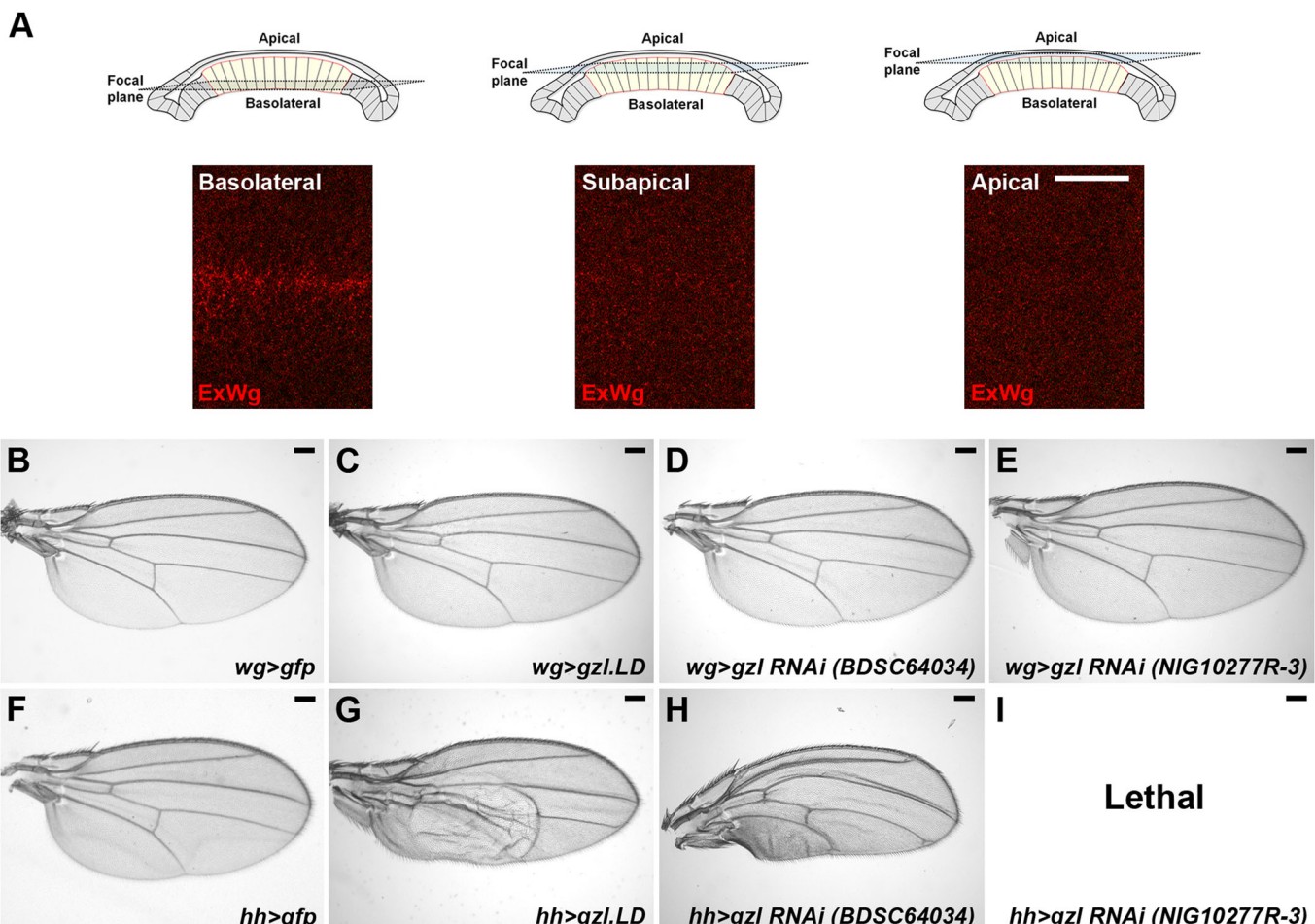

**Figure EV1.　Disruption of Gzl-mediated transcytosis has no impact on wing margin development.**

(**A**) The basolateral (left), subapical (middle), and apical (right) membrane domains of immunofluorescence staining of extracellular Wg (ExWg) for the wild-type wing imaginal discs are shown. (**B**) Shown is a wild-type adult wing expressing the *wg-Gal4* driver. (**C–E**) When a dominant-negative *gzl.LD* (**C**) or RNAi against *gzl* (targeting distinct regions of the *gzl* locus, **D** and **E**) was expressed using the *wg-Gal4* driver, no noticeable defects were observed in wing margin development. (**F**) Shown is a wild-type adult wing expressing the *hh-Gal4* driver. (**G–I**) When a dominant-negative *gzl.LD* (**G**) or RNAi against *gzl* (targeting distinct regions of the *gzl* locus, **H** and **I**) was expressed using the *hh-Gal4* driver, the resulting phenotypes include lethality in the adult fly (**I**) or disruptions in the development of the posterior compartment of the adult wing blade (**G** and **H**). Despite these effects, the wing margin remained unaffected, showing no apparent defects. Scale bars: (**A**) 25 μm; (**B–I**) 100 μm. Source data are available online for this figure.

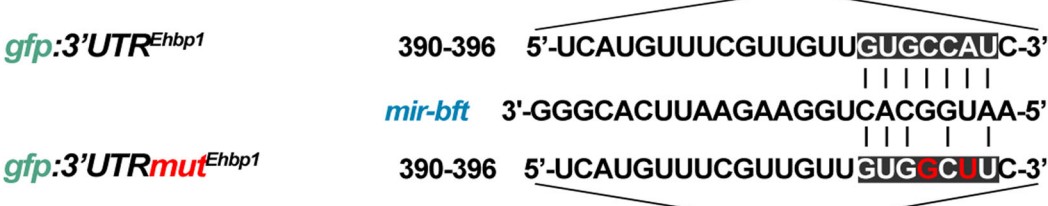

**A**

*gfp:3'UTR^Ehbp1*     390–396   5'-UCAUGUUUCGUUGUU**GUGCCAU**C-3'
                                               | | | | | | |
                        *mir-bft*   3'-GGGCACUUAAGAAGGUCACGGUAA-5'
                                               | | |   | |
*gfp:3'UTRmut^Ehbp1*   390–396   5'-UCAUGUUUCGUUGUU**GUGGCUU**C-3'

**Figure EV2.   *Ehbp1* is a bona-fide target of *mir-bft*.**

(**A**) Shown is the strategy for constructing the *mir-bft* GFP sensor. The 3'UTR of *Ehbp1* was cloned with GFP coding sequence to create the *tub:gfp:3'UTR^Ehbp1* sensor. The predicted target sequence for *mir-bft*, GUGCCAU, was mutated to GUGGCUU in the mutated *tub:gfp:3'UTRmut^Ehbp1* sensor. (**B–B″**) Upon overexpression of *mir-bft* using the *hh-Gal4* driver, a significant reduction in Ehbp1 was observed in the posterior compartment of the wing disc (**B, B′**), and a plot profile of immunofluorescence staining in (**B′**) was generated (**B″**) (for each genotype, *n* ≥ 3 biological replicates). Data are shown as mean ± SD. In these and all subsequent figures, wing discs are oriented with the anterior to the left and dorsal down, the anterior-posterior boundaries are indicated by dotted yellow lines. (**C–D″**) When *mir-bft* was overexpressed using the *hh-Gal4* driver, there was a reduction in the expression of *tub:gfp:3'UTR^Ehbp1* sensor in the posterior compartment of the wing disc (**C–C′**). In contrast, the expression levels of *tub:gfp:3'UTRmut^Ehbp1* sensor remained largely unchanged. Plot profiles of immunofluorescence staining in (**C′** and **D′**) were generated (**C″** and **D″**) (for each genotype, *n* ≥ 3 biological replicates). Data are shown as mean ± SD. Scale bars, 25 μm. Source data are available online for this figure.

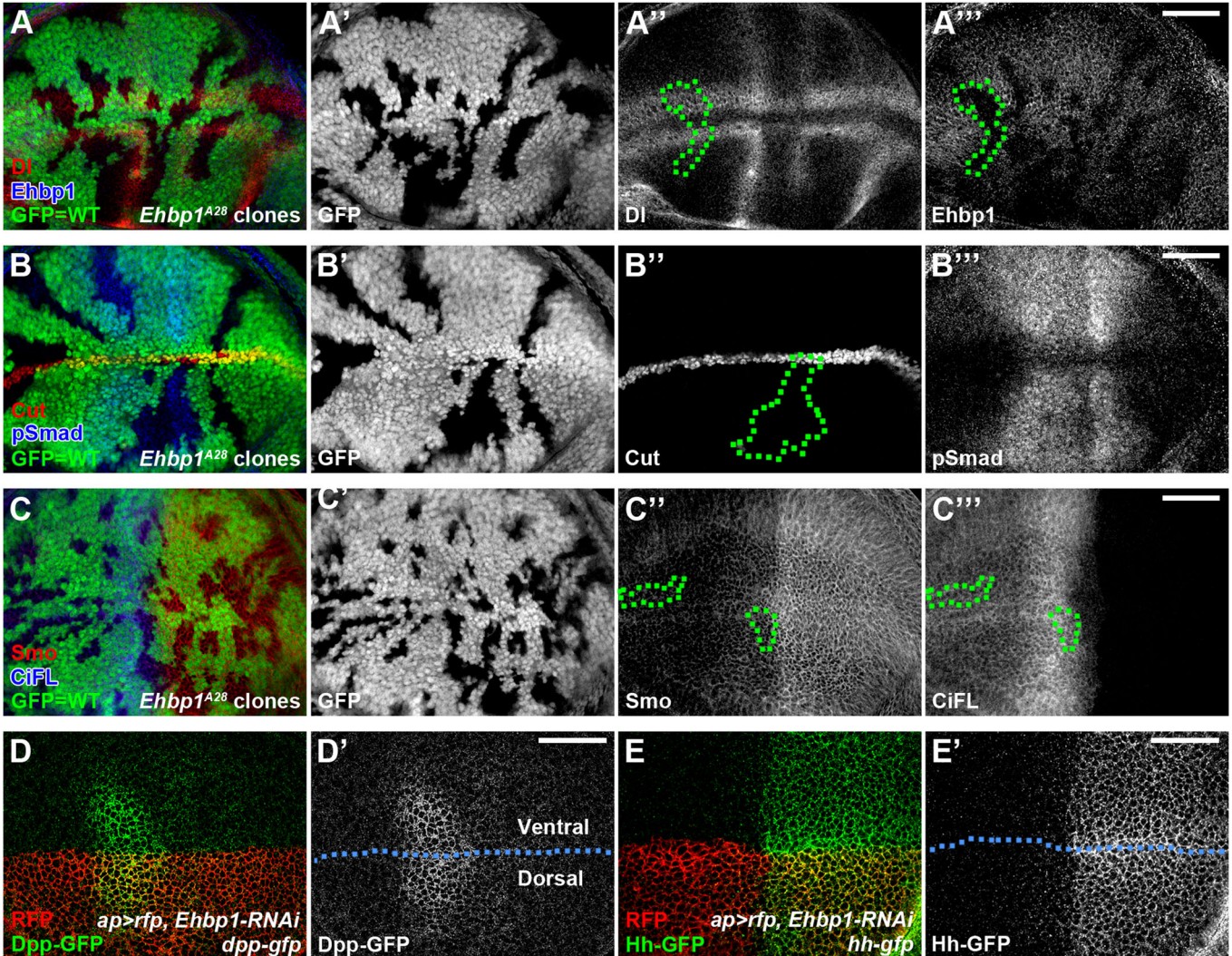

**Figure EV3. Ehbp1 does not play a role in the Notch, Dpp, or Hh signaling pathways during wing development.**

(A–C''') In negatively marked *Ehbp1^A28^* loss-of-function mutant clones (marked by the absence of GFP, outlined by dotted green lines), there were no obvious changes in the levels of the Notch signaling ligand Delta (Dl), the Notch signaling target Cut, the phosphorylation of the Dpp signaling activator Mothers against dpp (Mad/Smad), the Hh signaling activator Smoothened (Smo), or the activation of Hh signaling transcriptional factor Cubitus interruptus (CiFL). (D–E') When RNAi against *Ehbp1* was expressed in the dorsal compartment of the wing discs using the *apterous* (*ap*)-Gal4 driver, no alterations were observed in the distribution or levels of the morphogens Dpp or Hh, as indicated by GFP-trapped Dpp or Hh. The dotted blue lines indicate the D-V boundaries. Scale bars, 50 μm. Source data are available online for this figure.

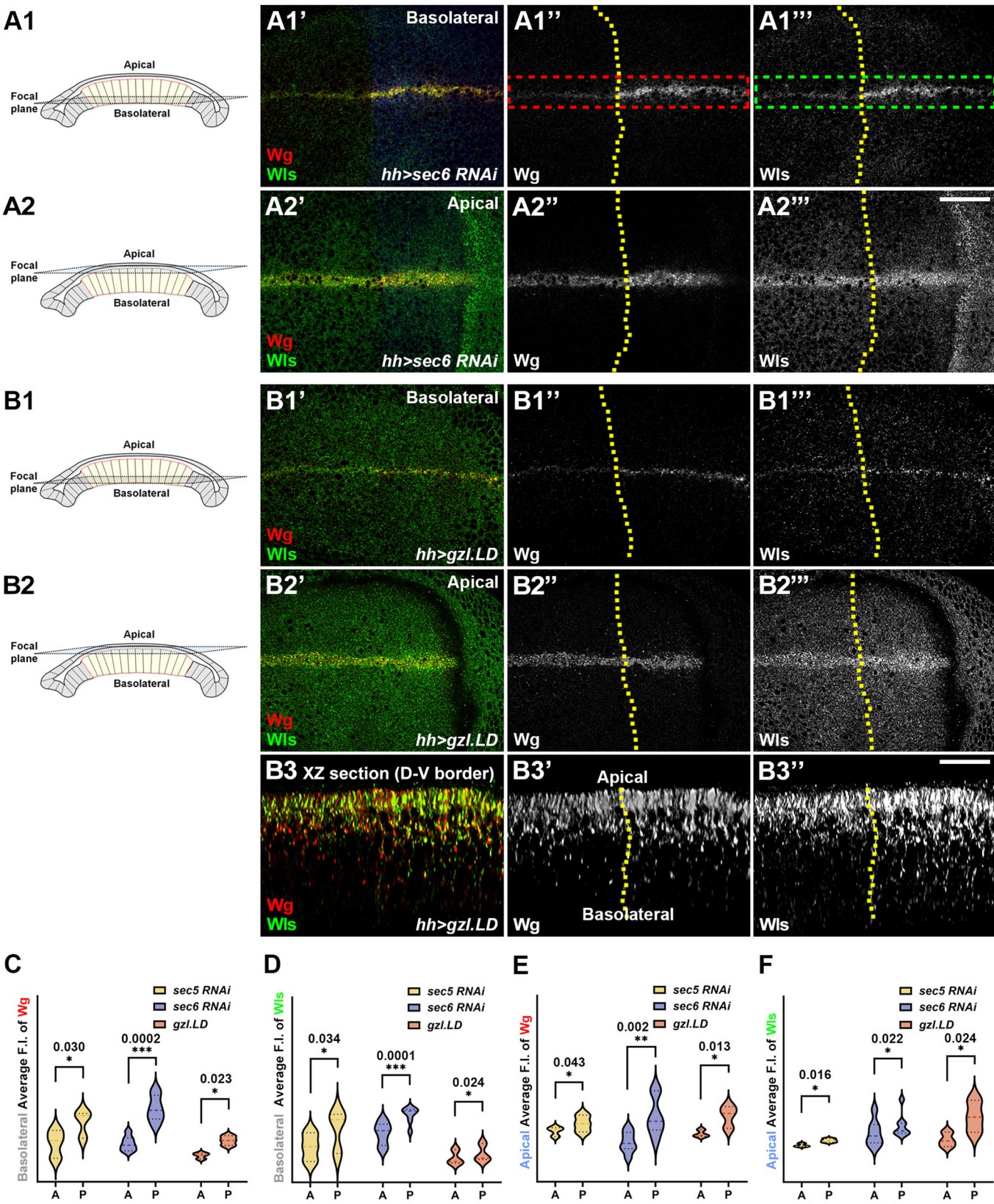

**Figure EV4. A transcytosis-independent pathway for basolateral Wg transport involves Wls activity.**

(**A1–B3″**) The basolateral (**A1–A1‴** and **B1–B1‴**) and apical (**A2–A2‴** and **B2–B2‴**) sections of immunofluorescence staining of Wg and Wls in the indicated genotypes are shown. When RNAi against *sec6* (**A1–A2‴**) or a dominant-negative *gzl.LD* (**B1–B2‴**) was expressed using the *hh-Gal4* driver, there was a significant accumulation of both Wls and Wg at the D-V boundary in the basolateral domains of the posterior compartments of the wing discs, while the apical accumulation was noticeably less intense. A 3D reconstruction of the D-V border cells (as viewed in an XZ section) from a wing disc expressing *gzl.LD* shows an enhanced transport of Wls to the basolateral domains (**B3–B3″**). Dotted yellow lines indicate the A-P boundaries. (**C–F**) Statistical analysis of Wg and Wls immunofluorescence intensity (F.I.) at the D-V boundary in both the basolateral and apical membrane domains was performed. The analyzed regions were delineated by rectangles, which were demarcated using red and green dashed lines. An example of this analysis is provided in (**A1–A1‴**). Immunofluorescence intensity data are presented as violin plots (for each genotype, $n \geq 3$ biological replicates). Two-tailed Student's t-tests were employed to analyze the differences between anterior and posterior F.I. *$p < 0.05$. **$p < 0.01$. ***$p < 0.001$. Scale bars, 25 μm. Source data are available online for this figure.

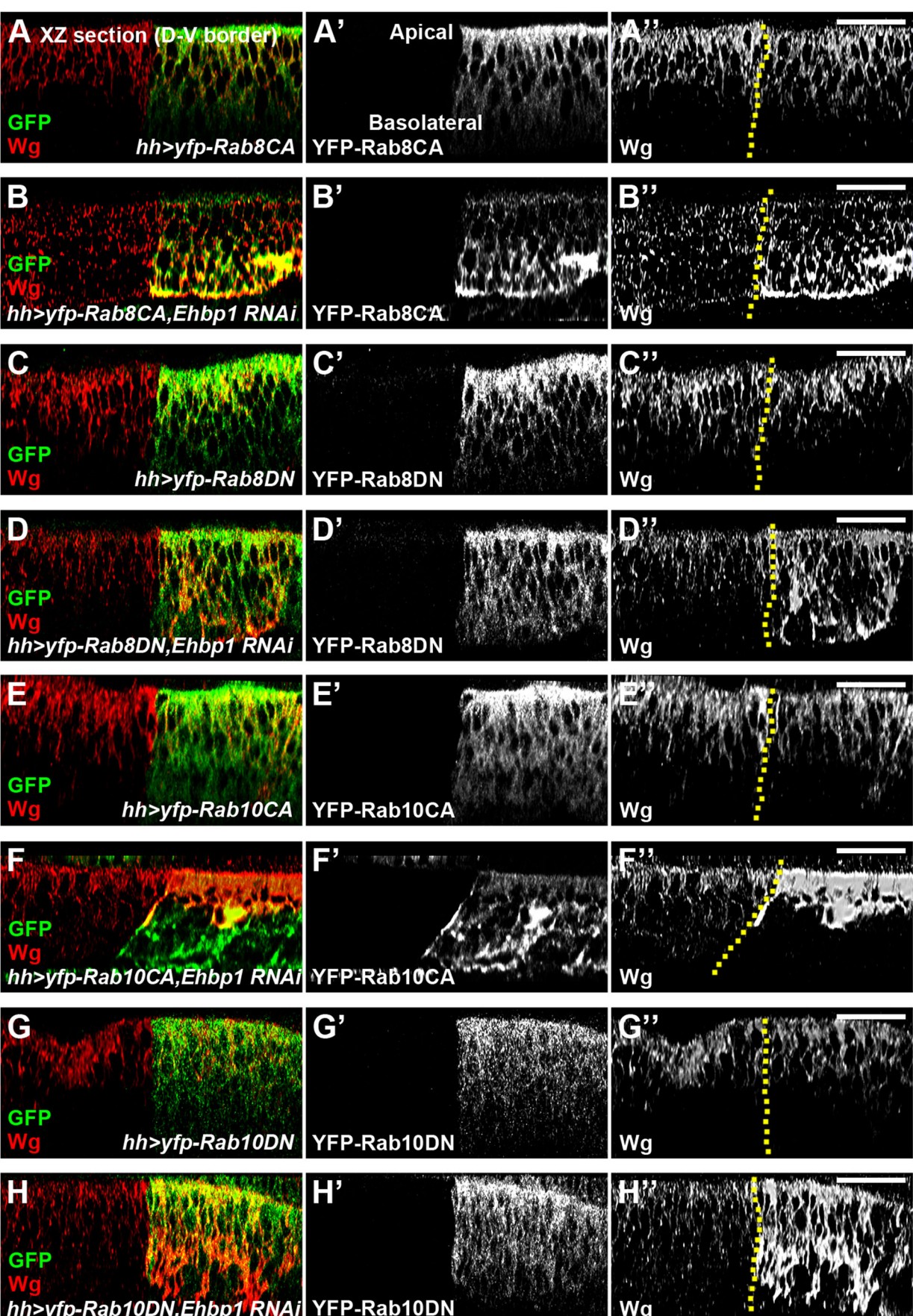

**Figure EV5.  Neither Rab8 nor Rab10 is essential for the intracellular transport of Wg.**

3D reconstructions of the D-V sections of the wing discs for the indicated genotypes are shown. Dotted yellow lines indicate the A-P boundaries. (**A–A″**) In cells where the constitutively active (CA) form of *Rab8* (*yfp-Rab8CA*) was overexpressed using the *hh-Gal4* driver, YFP-Rab8CA predominantly localized to the apical domain (**A′**). This did not appear to disrupt the polarized distribution of Wg (**A″**). (**B–B″**) When *Ehbp1* RNAi and *yfp-Rab8CA* were expressed using the *hh-Gal4* driver, YFP-Rab8CA was mainly found in the basolateral domain (**B′**), and Wg accumulated in the same domain. (**C–C″**) Overexpression of the dominant negative (DN) form of *Rab8* (*yfp-Rab8DN*) using the *hh-Gal4* driver resulted in YFP-Rab8DN being predominantly localized to the apical domain (**C′**), with no obvious defects in the polarized distribution of Wg (**C″**). (**D–D″**) Co-expression of *Ehbp1* RNAi and *yfp-Rab8DN* using the *hh-Gal4* driver showed no obvious defects in the distribution of YFP-Rab8DN (**D′**), although Wg accumulated in the basolateral domain (**D″**). (**E–E″**) The constitutively active (CA) form of *Rab10* (*yfp-Rab10CA*) overexpressed using the *hh-Gal4* driver was predominantly localized to the apical domain (**E′**), with no obvious impact on the polarized distribution of Wg (**E″**). (**F–F″**) When both *Ehbp1* RNAi and *yfp-Rab10CA* were expressed using the *hh-Gal4* driver, YFP-Rab10CA was mainly found in the basolateral domain (**F′**), while Wg accumulated in the apical domain (**F″**). (**G–G″**) When the dominant negative (DN) form of *Rab10* (*yfp-Rab8DN*) was overexpressed using the *hh-Gal4* driver, YFP-Rab10DN predominantly localized to the apical domain (**G′**), with no obvious defects observed in the polarized distribution of Wg (**G″**). (**H–H″**) Co-expression of *Ehbp1* RNAi and *yfp-Rab10DN* using the *hh-Gal4* driver showed no obvious defects in the distribution of YFP-Rab10DN (**H′**), although Wg accumulated in the basolateral domain (**H″**). Scale bars, 25 μm. Source data are available online for this figure.

