## [Peer Review File · EMBO Reports]

Ehbp1 orchestrates orderly sorting of Wnt/Wingless to the basolateral and apical cell membranes

Yuan Gao, Jing Feng, Yansong Zhang, Mengyuan Yi, Lebing Zhang, Yan Yan, Alan Zhu, and Min Liu

Corresponding author(s): Alan Zhu (zhua@pku.edu.cn) , Min Liu (liumin02@pku.edu.cn)

Review Timeline:

Submission Date:	15th Apr 24
Editorial Decision:	29th May 24
Revision Received:	24th Aug 24
Editorial Decision:	16th Sep 24
Revision Received:	17th Sep 24
Accepted:	27th Sep 24

Editor: Deniz Senyilmaz Tiebe / Martina Rembold

Transaction Report:

Dear Dr. Zhu,

Thank you for submitting your manuscript to EMBO Reports. My apologies for this unusual delay in getting back to you. Three referees agreed to review your manuscript. So far, we have received two referee reports that are copied below. Given that both referees are in fair agreement that you should be given a chance to revise the manuscript, I would like to ask you to begin revising your study along the lines suggested by the referees.

Please note that this is a preliminary decision made in the interest of time, and that it is subject to change should the third referee offer very strong and convincing reasons for this. As soon as we receive the final report on your manuscript, we will forward it to you as well.

Referees express interest in the proposed role of Ehbp1 in regulation of Wnt/Wingless sorting. However, they also raise some concerns that need to be addressed to consider publication here. We would like to encourage you to address the concerns of #1 regarding the mammalian conservation experimentally as we believe that it will significantly strengthen the manuscript. Please contact me if you would like to discuss this point further.

Given these positive recommendations, we would like to invite you to revise your manuscript with the understanding that the referee concerns (as in their reports) must be fully addressed and their suggestions taken on board. Please address all referee concerns in a complete point-by-point response. Acceptance of the manuscript will depend on a positive outcome of a second round of review. It is EMBO reports policy to allow a single round of major experimental revision only and acceptance or rejection of the manuscript will therefore depend on the completeness of your responses included in the next, final version of the manuscript.

We realize that it is difficult to revise to a specific deadline. In the interest of protecting the conceptual advance provided by the work, we recommend a revision within 3 months. Please discuss the revision progress ahead of this time with me if you require more time to complete the revisions, or if you have questions or comments regarding the revision (also by video chat).

1. A data availability section providing access to data deposited in public databases is missing (where applicable).
2. Your manuscript contains statistics and error bars based on $n=2$. Please use scatter plots in these cases.

You can submit the revision either as a Scientific Report or as a Research Article. For Scientific Reports, the revised manuscript can contain up to 5 main figures and 5 Expanded View figures, and it should not exceed 27000 characters. If the revision leads to a manuscript with more than 5 main figures it will be published as a Research Article. In this case the Results and Discussion section should be separate. If a Scientific Report is submitted, these sections have to be combined. This will help to shorten the manuscript text by eliminating some redundancy that is inevitable when discussing the same experiments twice. In either case, all materials and methods should be included in the main manuscript file.

4) a .docx formatted letter INCLUDING the reviewers' reports and your detailed point-by-point responses to their comments. As part of the EMBO publication's Transparent Editorial Process, EMBO reports publishes online a Review Process File (RPF) to accompany accepted manuscripts. This File will be published in conjunction with your paper and will include the referee reports, your point-by-point response and all pertinent correspondence relating to the manuscript.

<https://www.embopress.org/page/journal/14693178/authorguide#transparentprocess>

5) a complete author checklist, which you can download from our author guidelines

<https://www.embopress.org/page/journal/14693178/authorguide>. Please insert information in the checklist that is also reflected in the manuscript. The completed author checklist will also be part of the RPF.

6) Please note that all corresponding authors are required to supply an ORCID ID for their name upon submission of a revised manuscript (<<https://orcid.org/>>). Please find instructions on how to link your ORCID ID to your account in our manuscript tracking system in our Author guidelines

<<https://www.embopress.org/page/journal/14693178/authorguide#authorshipguidelines>>

7) Before submitting your revision, primary datasets produced in this study need to be deposited in an appropriate public database (see <https://www.embopress.org/page/journal/14693178/authorguide#datadeposition>). Please remember to provide a reviewer password if the datasets are not yet public. The accession numbers and database should be listed in a formal "Data Availability" section placed after Materials & Method (see also

<https://www.embopress.org/page/journal/14693178/authorguide#datadeposition>). Please note that the Data Availability Section is restricted to new primary data that are part of this study. * Note - All links should resolve to a page where the data can be accessed. *

Additional information on source data and instruction on how to label the files are available:

<https://www.embopress.org/page/journal/14693178/authorguide#sourcedata>

9) Our journal encourages inclusion of *data citations in the reference list* to directly cite datasets that were re-used and obtained from public databases. Data citations in the article text are distinct from normal bibliographical citations and should directly link to the database records from which the data can be accessed. In the main text, data citations are formatted as follows: "Data ref: Smith et al, 2001" or "Data ref: NCBI Sequence Read Archive PRJNA342805, 2017". In the Reference list, data citations must be labeled with "[DATASET]". A data reference must provide the database name, accession number/identifiers and a resolvable link to the landing page from which the data can be accessed at the end of the reference. Further instructions are available at <http://www.embopress.org/page/journal/14693178/authorguide#referencesformat>

10) Regarding data quantification (see Figure Legends:

<https://www.embopress.org/page/journal/14693178/authorguide#figureformat>)

12) Please also note our reference format:

13) All Materials and Methods need to be described in the main text. We would encourage you to use 'Structured Methods', our new Methods format. According to this format, the Methods section should include a Reagents and Tools Table (listing key reagents, experimental models, software and relevant equipment and including their sources and relevant identifiers) followed by a Methods and Protocols section in which we encourage the authors to describe their methods using a step-by-step protocol format with bullet points, to facilitate the adoption of the methodologies across labs. More information on how to adhere to this format as well as downloadable templates (.doc or .xls) for the Reagents and Tools Table can be found in our author guidelines: <https://www.embopress.org/page/journal/14693178/authorguide#manuscriptpreparation>.

I look forward to seeing a revised version of your manuscript when it is ready. Please let me know if you have questions or comments regarding the revision.

Kind regards,

Deniz Senyilmaz Tiebe

Deniz Senyilmaz Tiebe, PhD
Scientific Editor
EMBO Reports

Referee #1:

comments to the authors:

In this manuscript, Gao and colleagues aimed to elucidate novel regulators of Wg intracellular transport (apical vs. basolateral). In an initial miRNA screen in the *Drosophila melanogaster* larval wing disc they identified mir-bft, the overexpression of which resulted in an increase of intracellular accumulation of Wg and a reduction of Wg secretion at the apical cell surface. The knockdown of Ehbp1, a predicted mir-bft target, recapitulated these observations. The authors then use a variety of tools to convincingly uncover the molecular mechanism behind their observation. They propose a mechanism wherein Wg (in association with Wls) is usually directed towards (and secreted at) the basolateral side, governed through interaction with the AP-1 complex. However, Ehbp1 can compete with AP-1 for Wls binding, as a result of which, Wg (in association with Wls) is secreted apically.

Despite a lot being known about Wg/Wnt-ligand transport (especially intercellular), this is an important manuscript shedding light on the mechanisms behind the less studied intracellular transport of Wg/Wnt ligands. The extracellular Wg imaging data in the *Drosophila* wing disc provides strong support for the proposed mechanism. The weak part of the manuscript lies in the claimed conservation of this mechanism in mammals. I suggest either removing this part, as the findings in the *Drosophila* wing disc are strong enough to be published on their own or add further supporting data, as suggested below to strengthen the claim.

Major comments:

- The authors chose to study, amongst other readouts, the expression of the Wnt target gene *senseless* to analyze whether differential intracellular transport and secretion of Wg has a functional impact on Wnt signaling. However, they only show the loss of *senseless* (in *ehbp1* knockdown clones) in the posterior wing disc. Since posterior loss of *senseless* expression could also be a result of slightly younger wing disc, it would be ideal to further support this data with images showing clones on the anterior side of the wing disc.

- The authors chose to probe the mammalian conservation of the proposed intracellular Wg/Wnt-ligand transport mechanism using MDCK cells. However, the data provided is quite minimalistic, comprising a single Streptavidin-based CO-IP experiment to study the interaction of WNT1 and EHBP1. There are multiple open questions that should be addressed here if the authors wish to make the claim of mammalian conservation of this mechanism. While the biotin/streptavidin-based CO-IP system is quite elegant, the authors cannot exclude that laterally secreted WNT1 is biotinylated by the apically provided Sulfo-NHS-LC-biotin medium. It would therefore be more convincing if the authors would add imaging data to show the differential intracellular localization and secretion of WNT1, which should be easy to do given their use of a WNT1-GFP fusion protein. In that context, the usage of WNT1-GFP overexpression poses two problems that the authors should address: 1) the strong overexpression of WNT1 might actually alter how it is intracellularly transported and 2) the addition of a GFP tag to WNT1 might alter how it is intracellularly transported. An approach similar to the *Drosophila* wing disc experiments might therefore be more convincing, namely A) the study and imaging of a WNT ligand that is endogenously expressed in MDCK cells + extracellular antibody staining or B) the study and imaging of an endogenously tagged version of said WNT ligand. Furthermore, the authors should also show that the other elements of their proposed mechanism are conserved in the mammalian system e.g., by knocking out/down WLS and/or AP-1 in the MDCK cells in combination with the EHBP1 knockdown and analyzing WNT-ligand localization/secretion in these conditions.

Minor comments:

- The authors convincingly show an increase in basolateral extracellular Wg in the ehbp1 knockdown condition. However, the concomitant reduction of Wg at the apical surface is less convincing. The authors should remove this statement or repeat the experiment to display this effect (apical reduction) more convincingly.
- Although the extracellular antibody staining of Wg is of high quality, repetition of some experimental conditions (e.g., ehbp1 knockdown) using an endogenously tagged Wg-GFP (e.g., 10.1073/pnas.1405500111) could provide an even better visualization of especially the intracellular distribution of Wg, helping to further substantiate the authors' claims.
- The authors focus on Wg (in *Drosophila*) and WNT1 (in mammals) for their experimental elucidation and validation of the proposed intracellular transport mechanism. It would be nice if they could discuss (or even proof with additional experiments) whether they believe this mechanism also holds true for other Wnt family ligands.

Referee #2:

This manuscript presents compelling data that show a role for Ehbp1 in the secretory pathways for Wg/Wnt. This is a very important result and will help to resolve some of the controversies about apical/basolateral sorting of Wnts, both in the fly and in the vertebrate systems.

The writing is clear and concise; every question that crossed my mind as I read the text was answered in the next figure. The figures are well-organized and they beautifully illustrate the main points of the paper. The authors are meticulous in their presentation, with attention to even small details such as putting white outlines around the darker-hued labels in fluorescent images.

The authors are very thorough in their analyses, using both loss of function mutations (in somatic clones) and RNA interference knockdown to establish function, as well as ectopic expression to show an opposite effect. They provide strong co-immunoprecipitation data indicating competition of Wntless and Ehbp1 for the AP-1 adaptor. I found no weaknesses in this paper (highly unusual for this reviewer) and can only offer one suggestion for improvement: in Fig. 3A3 - change the clone labels within the image from A28/A28 and A28/+ to -/- and +/-, so that the labelling is consistent with other figures.

Ehbp1 orchestrates orderly sorting of Wnt/Wingless to the basolateral and apical cell membranes (Manuscript# EMBOR-2024-59424)**Responses to Reviewer #1's comments****(Reviewers' comments in italic)**

In this manuscript, Gao and colleagues aimed to elucidate novel regulators of Wg intracellular transport (apical vs. basolateral). In an initial miRNA screen in the Drosophila melanogaster larval wing disc they identified mir-bft, the overexpression of which resulted in an increase of intracellular accumulation of Wg and a reduction of Wg secretion at the apical cell surface. The knockdown of Ehbp1, a predicted mir-bft target, recapitulated these observations. The authors then use a variety of tools to convincingly uncover the molecular mechanism behind their observation. They propose a mechanism wherein Wg (in association with Wls) is usually directed towards (and secreted at) the basolateral side, governed through interaction with the AP-1 complex. However, Ehbp1 can compete with AP-1 for Wls binding, as a result of which, Wg (in association with Wls) is secreted apically.

Despite a lot being known about Wg/Wnt-ligand transport (especially intercellular), this is an important manuscript shedding light on the mechanisms behind the less studied intracellular transport of Wg/Wnt ligands. The extracellular Wg imaging data in the Drosophila wing disc provides strong support for the proposed mechanism. The weak part of the manuscript lies in the claimed conservation of this mechanism in mammals. I suggest either removing this part, as the findings in the Drosophila wing disc are strong enough to be published on their own or add further supporting data, as suggested below to strengthen the claim.

Response: We thank the reviewer for the positive feedback on our work. We have followed his/her suggestions to improve our manuscript.

Major comments:

1. The authors chose to study, amongst other readouts, the expression of the Wnt target gene senseless to analyze whether differential intracellular transport and secretion of Wg has a functional impact on Wnt signaling. However, they only show the loss of senseless (in ehpb1 knockdown clones) in the posterior wing disc. Since posterior loss of senseless expression could also be a result of slightly younger wing disc, it would be ideal to further support this data with images showing clones on the anterior side of the wing disc.

Response: We have followed the reviewer's advice and replaced **Figure 2B-B''** with images showing the Senseless staining in *Ehpb1* clones on the anterior side of the wing disc.

2. The authors chose to probe the mammalian conservation of the proposed intracellular Wg/Wnt-ligand transport mechanism using MDCK cells. However, the data provided is quite minimalistic, comprising a single Streptavidin-based CO-IP experiment to study the interaction of WNT1 and EHBP1. There are multiple open questions that should be addressed here if the authors wish to make the claim of mammalian conservation of this mechanism. While the biotin/streptavidin-based CO-IP system is quite elegant, the authors cannot exclude that laterally secreted WNT1 is biotinylated by the apically provided Sulfo-NHS-LC-biotin medium. It would therefore be more convincing if the authors would add imaging data to show the differential intracellular localization and secretion of WNT1, which should be easy to do given their use of a WNT1-GFP fusion protein. In that context, the usage of WNT1-GFP overexpression poses two problems that the authors should address: 1) the strong overexpression of WNT1 might actually alter how it is intracellularly transported and 2) the addition of a GFP tag to WNT1 might alter how it is intracellularly transported. An approach similar to the Drosophila wing disc experiments might therefore be more convincing, namely A) the study and imaging of a WNT ligand that is endogenously expressed in MDCK cells + extracellular antibody staining or B) the

study and imaging of an endogenously tagged version of said WNT ligand. Furthermore, the authors should also show that the other elements of their proposed mechanism are conserved in the mammalian system e.g., by knocking out/down WLS and/or AP-1 in the MDCK cells in combination with the EHBP1 knockdown and analyzing WNT-ligand localization/secretion in these conditions.

Response: We are grateful for the reviewer's insightful suggestions. In response, we conducted a comprehensive analysis of WNT family gene expression using publicly available RNA-seq data from MDCK cells ([10.1186/s12864-015-2036-9](https://doi.org/10.1186/s12864-015-2036-9) and [10.3389/fphys.2017.00997](https://doi.org/10.3389/fphys.2017.00997)). Subsequently, we validated the expression of *WNT2*, *WNT5A*, *WNT5B*, *WNT7A*, *WNT7B*, *WNT9A*, and *WNT11* genes in MDCK cells using RT-PCR (**Response Figure 1**). To study the intracellular localization of WNT family proteins in MDCK cells, we tested various antibodies: rabbit anti-WNT2 (Abclonal, A13562), rabbit anti-WNT5B (Abclonal, A8313), rabbit anti-WNT7A (Proteintech, 27177-1-AP), rabbit anti-WNT7B (Abclonal, A17004), and rabbit anti-WNT9A (Abclonal, A7939). Among these, the WNT7A antibody effectively recognized endogenous WNT7A protein. Using this antibody, we demonstrated that the intracellular trafficking of WNT7A is regulated by EHBP1, WLS, and AP1 (**Figure 6B-M**). This regulatory mechanism is consistent with our findings in *Drosophila*, suggesting the conservation of this pathway across different species.

Response Figure 1. RT-PCR analysis of WNT family gene expression in MDCK cells. Transcripts of *WNT2*, *WNT5A*, *WNT5B*, *WNT7A*, *WNT7B*, *WNT9A*, and *WNT11* were detected in cDNA samples prepared from MDCK cells.

Minor comments:

- The authors convincingly show an increase in basolateral extracellular Wg in the *ehbp1* knockdown condition. However, the concomitant reduction of Wg at the apical surface is less convincing. The authors should remove this statement or repeat the experiment to display this effect (apical reduction) more convincingly.

Response: We have followed the reviewer's recommendation and removed the aforementioned statement.

- Although the extracellular antibody staining of Wg is of high quality, repetition of some experimental conditions (e.g., *ehbp1* knockdown) using an endogenously tagged Wg-GFP (e.g., [10.1073/pnas.1405500111](https://doi.org/10.1073/pnas.1405500111)) could provide an even better visualization of especially the intracellular distribution of Wg, helping to further substantiate the authors' claims.

Response: We have followed the reviewer's suggestion and examined the intracellular distribution of Wg-GFP following *Ehbp1* knockdown. The findings of this experiment are presented in **Appendix Figure S1**.

- The authors focus on Wg (in *Drosophila*) and WNT1 (in mammals) for their experimental elucidation and validation of the proposed intracellular transport mechanism. It would be nice if they could discuss (or even proof with additional experiments) whether they believe this mechanism also holds true for other Wnt family ligands.

Response: This issue has been addressed in the Discussion section of the revised manuscript.

Response to Reviewer #2's comments

(Reviewer's comments in italic)

This manuscript presents compelling data that show a role for Ehbp1 in the secretory pathways for Wg/Wnt. This is a very important result and will help to resolve some of the controversies about apical/basolateral sorting of Wnts, both in the fly and in the vertebrate systems.

The writing is clear and concise; every question that crossed my mind as I read the text was answered in the next figure. The figures are well-organized and they beautifully illustrate the main points of the paper. The authors are meticulous in their presentation, with attention to even small details such as putting white outlines around the darker-hued labels in fluorescent images.

The authors are very thorough in their analyses, using both loss of function mutations (in somatic clones) and RNA interference knockdown to establish function, as well as ectopic expression to show an opposite effect. They provide strong co-immunoprecipitation data indicating competition of Wntless and Ehbp1 for the AP-1 adaptor. I found no weaknesses in this paper (highly unusual for this reviewer) and can only offer one suggestion for improvement: in Fig. 3A3 - change the clone labels within the image from A28/A28 and A28/+ to -/- and +/-, so that the labelling is consistent with other figures.

Response: We greatly appreciate the reviewer's positive comments and have followed his/her suggestion to change the clone labels in **Fig. 3A3**.

Dear Dr. Zhu,

Thank you for submitting your revised manuscript. It has now been seen by one of the original referees.

As you can see, the referee finds that the study is significantly improved during revision and recommends publication. However, I need you to address the points below before I can accept the manuscript.

- Please add a "Disclosure Statement and Competing Interests" title before the following sentence "The authors declare no competing interests."
- Please remove 'Author Contributions' section from the manuscript text.
- We note that the funding information is currently not congruent - the following is missing from the manuscript tracking system:
 - Qidong-SLS Innovation Fund
 - the Peking-Tsinghua Center for Life Sciences
 - the Ministry of Education Key Laboratory of Cell Proliferation and Differentiation
 - Peking University President's Scholarship Awardees
 - Boya Postdoctoral Fellowship
 - BDSC and DGRC, supported by the grants from National Institutes of Health P40OD018537 and 2P40OD010949
- We note that Fig. 6H-I panels are currently not called out in the text.
- We note the following regarding the Appendix file: A Table of Contents needs to be added which has page numbers on the title page. The nomenclature of Appendix tables needs to be corrected to Appendix Table S1, Appendix Table S2 and also needs to be updated in the manuscript text.
- The Reagents&Tools table needs to be uploaded separately by choosing the relevant file type.
- Source data for the EV figures should be grouped into one zip folder.
- Our production/data editors have asked you to clarify several points in the figure legends:
 - o Please note that the scale bar information in the legend of figure EV 1b-i is mislabeled as figure EV 1b-e in the manuscript. This needs to be rectified.
 - o Please note that the legend for figure EV 4b3' is mislabeled as figure EV 4b3"" in the manuscript. This needs to be rectified.
 - o Please note that the exact p values are not provided in the legends of figures 1i; 2d; 3e; 4h-i; EV 4c-f.
 - o Please note that the scale bar is missing for figures 1g, h1, h2; 2a; 3b-c; 4a-b, d, f.
 - o Please note that the scale bar needs to be defined for figures 1d-f.
 - o Please note that the "+/-", "-/-" is not defined in the legend of figure 2a, b; 3a1, a2, a3. This needs to be rectified.
 - o Please note that the green dotted lines are not defined in the legend of figures 3a1', a2', a3', a3"", a3""; 4a', b'c1', c2', f1. This needs to be rectified.
 - o Please note that the yellow dotted lines are not defined in the legend of figures 3b', c', d'; 4f1', f2', g1', g2'; 5a1', a2'; EV 2b', c', d'; EV 5a", b', c", d", e", f", g", h". This needs to be rectified.
- Papers published in EMBO Reports include a 'synopsis' and 'bullet points' to further enhance discoverability. Both are displayed on the html version of the paper and are freely accessible to all readers. The synopsis includes a short standfirst summarizing the study in 1 or 2 sentences (max 35 words) that summarize the paper and are provided by the authors and streamlined by the handling editor. I would therefore ask you to include your synopsis blurb and 3-5 bullet points listing the key experimental findings.
- In addition, please provide an image for the synopsis. This image should provide a rapid overview of the question addressed in the study but still needs to be kept fairly modest since the image size cannot exceed 550 (width) x 300-600 (height) pixels.

Thank you again for giving us to consider your manuscript for EMBO Reports, I look forward to your minor revision.

Kind regards,

Deniz Senyilmaz Tiebe

--

Deniz Senyilmaz Tiebe, PhD
Senior Scientific Editor
EMBO Reports

Referee #1:

Re-review:

The authors addressed all major concerns.

Major concern 1: The expression of the Wnt target gene (senseless) was chosen to analyze whether differential intracellular transport and secretion of Wg has a functional impact on Wnt signaling. Previously, the authors only showed the loss of senseless expression (in ehpb1 knockdown clones) in the posterior wing disc. Since posterior loss of senseless expression

could also be a result of slightly younger wing disc, I requested for better wing disc images. The wing discs were replaced with suitable and convincing alternatives showing loss of senseless in the ehpb1 knockdown clones in anterior regions of the disc.

Major concern 2: The authors chose to probe the mammalian conservation of the proposed intracellular Wg/Wnt-ligand transport mechanism using MDCK cells. However, the data provided was minimalistic, comprising a single Streptavidin-based CO-IP experiment to study the interaction of WNT1 and EHBP1. The greatest concern was that Wnt1 is not endogenous to these cells, and the results could be an over expression/tagged gene artifact.

The authors addressed these concerns by performing an expression analysis of Wnts in MDCK cells, and found Wnt2, Wnt5a, Wnt5b, Wnt7a, Wnt7b, and Wnt9a were all expressed in these cells. They identified suitable Wnt7a/ExWnt7a antibodies to monitor apical basal localization in EHBP1, AP1, and Wls knockdown conditions. Although the staining was generally weak, it supported the results from the Drosophila wing disc experiments.

The authors have addressed all minor editorial requests.

Dr. Alan Zhu
Peking University
5 Yiheyuan Road
Beijing, Not Applicable 100871
China

Dear Dr. Zhu,

Since my colleague Deniz Senyilmaz Tiebe is currently traveling, I have temporarily taken over the handling of your manuscript. I have checked all files and am very pleased to accept your manuscript for publication in the next available issue of EMBO reports. Thank you for your contribution to our journal.

Yours sincerely,
